# Multi-Objective Molecular Design through Learning Latent Pareto Set

## Abstract

Molecular design inherently involves the optimization of multiple conflicting objectives, such as enhancing bio-activity and ensuring synthesizability. Evaluating these objectives often requires resource-intensive computations or physical experiments. Current molecular design methodologies typically approximate the Pareto set using a limited number of molecules. In this paper, we present an innovative approach, called Multi-Objective Molecular Design through Learning Latent Pareto Set (MLPS). MLPS initially utilizes an encoder-decoder model to seamlessly transform the discrete chemical space into a continuous latent space. We then employ local Bayesian optimization models to efficiently search for local optimal solutions (i.e., molecules) within predefined trust regions. Using surrogate objective values derived from these local models, we train a global Pareto set learning model to understand the mapping between direction vectors (called "preferences") in the objective space and the entire Pareto set in the continuous latent space. Both the global Pareto set learning model and local Bayesian optimization models collaborate to discover high-quality solutions and adapt the trust regions dynamically. Our work represents a novel approach to learning the mapping between preferences and the Pareto set in the latent space, specifically tailored for multi-objective molecular design, providing decision-makers with the capability to fine-tune their preferences and explore the Pareto set. Experimental results demonstrate that MLPS achieves state-of-the-art performance across various multi-objective scenarios, encompassing diverse objective types and varying numbers of objectives.

## 1 Introduction

Molecular design plays a pivotal role in a multitude of applications, including drug discovery (Meyers et al. (2021)), material science (Butler et al. (2018)), and catalyst development (Wan et al. (2020)). In this context, multi-objective molecular design (MMD) is a challenging endeavor. It aims to discover the Pareto set of molecules, where improving one objective inevitably entails compromising others.

Traditional approaches to MMD often involve simplifying multi-objective problems by converting them into single-objective ones using specific weightings (Abels et al. (2019); SV et al. (2022)). While effective to some extent, these methods rely on human experts for weighting. In general, weighting of multiple objectives is difficult, and inappropriate weighting leads to inappropriate solutions. Other techniques focus on identifying the Pareto set through a two-stage process: molecule sampling and non-dominated sorting (Yasonik (2020); Verhellen (2022)). However, this approach can become prohibitively expensive and time-consuming, especially in cases with multiple objectives and a large pool of molecules to consider. To improve sampling efficiency in MMD, Bayesian optimization has emerged as a powerful and efficient method (Xie et al. (2021); Gao et al. (2022)). Nevertheless, Bayesian optimization faces limitations when dealing with high-dimensional spaces and high computational complexity associated with Gaussian process inference. These challenges make it less suited for complex MMD tasks.

Conventional MMD approaches often yield a limited Pareto set (i.e., a small number of trade-off solutions), which may not align well with decision-makers' preferences. MMD problems may have a complicated Pareto set with various trade-offs. Accessing this entire set can provide significant ad-

vantages, enabling decision-makers to select the most preferred solution and accelerating the molecular design process. Recent advancements in this field, particularly Pareto set learning methods, aim to approximate the entire Pareto set using learning models (Lin et al. (2021; 2022)). However, most existing methods focus on low-dimensional Pareto sets, which are not suitable for addressing the complexity of high-dimensional and discrete MMD problems. These limitations underscore the need for a novel approach that can efficiently handle MMD scenarios while learning the comprehensive Pareto set.

In this paper, we introduce a groundbreaking approach to MMD, termed as Multi-Objective Molecular Design through Latent Pareto Set Learning (MLPS). The core of MLPS begins with the utilization of an encoder-decoder model, which effectively maps the discrete chemical space into a continuous latent space. This transformation process lays the foundation for subsequent operations. MLPS partitions the latent space into multiple trust regions and uses a local Bayesian optimization model for efficient sampling within each region. Notably, MLPS incorporates a global Pareto set learning model, which establishes a crucial link between direction vectors (i.e., preferences) in the objective space and the comprehensive Pareto set in the continuous latent space. It is capable of aggregating and disseminating valuable information from each trust region, thereby enhancing both global and local optimization processes. Our paper makes several significant contributions to the field of MMD: (The code of MLPS will be publicly accessible upon acceptance.)

- We introduce a novel approach that learns a mapping from a preference to the corresponding Pareto optimal solution in MMD. It provides a vast array of solutions for decision-makers to choose from, enabling them to select molecules that best align with their preferences and specific requirements. This is clearly different from existing MMD methods where preferences are often predefined. Our approach offers a fresh perspective on solving the problem.

- We develop an efficient information-sharing mechanism wherein local Bayesian models communicate with each other through a global neural network. This combination of local and global optimization enhances the search capability for the Pareto set, allowing for quicker convergence to better solutions.

- Our approach outperforms existing state-of-the-art methods across a wide range of multi-objective scenarios. This includes scenarios involving diverse types of objectives and varying numbers of objectives, showcasing its effectiveness and versatility.

## 2 RELATED WORKS

**Molecular Design.** Molecular design aims to improve specific molecular properties. Recent advances have seen the emergence of various artificial intelligence-based approaches. These approaches can be broadly categorized into two groups: 1) Generative Models: This category includes methods such as variational autoencoders (Lim et al. (2018); Liu et al. (2018); Jin et al. (2018a;b)), generative adversarial networks (Guimaraes et al. (2017); Kadurin et al. (2017); Prykhodko et al. (2019)), and diffusion models (Ho et al. (2020); Xu et al. (2021); Hoogeboom et al. (2022)). They work with continuous latent representations and require substantial data for training. 2) Combinatorial Optimization Methods: These methods involve combinatorial search in a discrete chemical space. They include evolutionary algorithms (A Nicolaou & Kannas (2012); Nigam et al. (2020); Ahn et al. (2020)), reinforcement learning (You et al. (2018); Jin et al. (2020); Fu et al. (2021a)), and Monte Carlo methods (Jensen (2019); Xie et al. (2021); Fu et al. (2021b); Sun et al. (2022)). While they do not require as much training data as generative models, they often rely on domain knowledge and can be expensive due to the evaluation of a large number of molecules. However, none of these existing methods can learn the mapping from preferences to the Pareto set for MMD. This limitation motivates the development of our MLPS, which aims to address this gap.

**Multi-Objective Bayesian Optimization.** Multi-objective Bayesian optimization has gained significant attention in recent years. Some techniques (Knowles (2006); Paria et al. (2020)) involve transforming a multi-objective problem into a set of single-objective ones by using scalarizing functions, allowing single-objective Bayesian optimization to be applied. Other approaches employ a Pareto domination method (Bradford et al. (2018)) or the hypervolume indicator (Daulton et al. (2020); Zhao et al. (2021)) to guide the optimization process. Bayesian optimization has

found applications in molecular design, which often involves learning a latent space from molecular data using generative models and then searching for molecules with desired properties in the latent space (Gómez-Bombarelli et al. (2018); Jin et al. (2018a); Tripp et al. (2020); Siivola et al. (2021)). However, existing Bayesian optimization methods in molecular design often face limitations. For example, transforming multi-objective problems into single-objective ones using weighted sums may not capture the full spectrum of solutions (i.e., multi-objective trade-offs). Additionally, representing complex chemical spaces in high-dimensional latent spaces can be challenging, and traditional multi-objective Bayesian optimization methods have limitations in handling high-dimensional spaces (Frazier (2018)). Our MLPS represents a pioneering effort in multi-objective molecular design with Bayesian optimization. It provides a novel framework to learn the mapping from preferences to the Pareto set while efficiently navigating the search in high-dimensional spaces.

**Trust Region.** Trust region methods (Conn et al. (2000); Powell et al. (2009)) aim to efficiently handle high-dimensional spaces by iteratively exploring trust regions. These methods simultaneously optimize multiple subspaces or regions in the search space. The key idea is to evaluate the improvement in objective functions within these trust regions and adjust the regions accordingly. Turbo (Eriksson et al. (2019)) is a recent work that combines the trust region methodology with Bayesian optimization. It introduces hyperrectangular trust regions, where each trust region has a center point and an edge length. The edge length can grow or shrink based on the performance of the trust region. If a trust region is deemed promising, its edge length is increased. Otherwise, the edge length is decreased. When the edge length becomes too small, the trust region is reinitialized. One limitation of Turbo is that it lacks communication among trust regions. Each trust region operates independently, which can lead to suboptimal exploration of the search space. This is where our MLPS differs. It incorporates a global model to facilitate information sharing among trust regions, which improves the overall search efficiency. Furthermore, we propose the use of the hypervolume indicator for the center setting and reinitialization strategy for trust regions, making our MLPS well-suited for MMD.

**Pareto Set Learning.** In the field of multi-objective optimization, there has been a growing interest in developing methods that learn to approximate the Pareto set, such as a reinforcement learning-based approach (Parisi et al. (2016)) and an incremental learning-based approach (Liu et al. (2021)). A prevalent trend in multi-objective optimization involves integrating preference information into neural networks. This approach has demonstrated success in various domains, including multi-task learning (Sener & Koltun (2018); Navon et al. (2020)) and reinforcement learning (Abdolmaleki et al. (2020; 2021)). By incorporating preferences, neural networks can adapt to specific preferences of decision-makers. Some notable works have aimed to learn and model the entire Pareto set. For instance, P-MOCO (Lin et al. (2021)) introduces an end-to-end reinforcement learning algorithm designed to train models capable of accommodating different preferences in multi-objective combinatorial optimization problems. In the context of expensive multi-objective optimization problems, PSL (Lin et al. (2022)) has been developed. However, it is primarily designed for low-dimensional problems. Moreover, PSL rely on a single global model, which may not be well-suited for capturing highly complicated Pareto sets encountered in molecular design. P-MOCO might have similar issues. In contrast, our MLPS takes a hybrid approach that combines several local models with a global model. This hybrid architecture is specifically designed to address the challenges posed by complicated high-dimensional Pareto sets in molecular design.

## 3 METHOD

In our study, we treat the multi-objective molecular design problem as a multi-objective maximization problem, formulated as follows:

$$\max_{\boldsymbol{x} \in \boldsymbol{\chi}} \mathbf{F}(\boldsymbol{x}) = [f_1(\boldsymbol{x}), f_2(\boldsymbol{x}), \cdots, f_M(\boldsymbol{x})], \quad (1)$$

Here, $\boldsymbol{x}$ denotes a solution (i.e., molecule) in the $N$-dimensional latent space $\boldsymbol{\chi} \in \mathbb{R}^N$, $M$ represents the number of objectives, and evaluating these objective functions can be expensive.

In this section, we first provide an overview of our MLPS framework in Subsection 3.1. Then, we delve into the specifics of the local and global models within our MLPS in Subsection 3.2 and Subsection 3.3, respectively. To provide readers with a solid foundation in multi-objective optimization, Appendix A offers explanations on fundamental concepts such as Pareto dominance, Pareto set/front, the hypervolume (HV) indicator ($f_{HV}$), the hypervolume contribution ($f_{HVC}$), and the

hypervolume improvement ($f_{HVI}$). For those seeking more in-depth information, Appendix B supplies additional details about our work.

## 3.1 FRAMEWORK

The framework of our MLPS is illustrated in Fig 1. MLPS operates as a guided search model, where guidance and feedback from current candidates continuously inform and improve the search strategy in each iteration. This search is conducted within a continuous latent space established by a pre-trained encoder-decoder model. We use the SELFIES variational autoencoder with a latent space dimensionality of 256 (Maus et al. (2022)) for all molecular design tasks in this work. Details about the molecular representations and the encoder-decoder model are found in Appendix B.1.

The first step in the search process involves setting the initial molecular embeddings. MLPS offers two methods for accomplishing this: 1) One method samples a set of molecules from a database, and their embeddings are obtained using the encoder. 2) The other method directly samples initial embeddings from the latent space. In our work, we employ the Sobol sampler (Renardy et al. (2021)), which enables uniform sampling in the high-dimensional latent space.

Once the initial molecular embeddings are in place, we proceed to initialize multiple trust regions within the latent space. Within each trust region, we build a local surrogate model. Subsequently, based on these local surrogate models, we train a global Pareto set learning model, which is implemented as a neural network. The global Pareto set learning model is designed to establish a mapping between direction vectors (i.e., preferences) in the objective space and the entire Pareto set across the latent space. To train the global Pareto set learning model, we randomly sample a set of preferences and utilize it to predict Pareto optimal solutions. The predicted solutions are then evaluated by the local surrogate models within their respective trust regions to generate loss values. The loss is utilized as the training signal for the global model, enabling it to improve its mapping accuracy.

After training the global model in the current iteration, we employ it to generate predicted Pareto optimal solutions by randomly inputting preferences into the model. Subsequently, we select the best global batch of solutions $\boldsymbol{X}_g$ based on their hypervolume improvement values. Additionally, we sample a set of solutions within the trust regions, and we select the best local batch of solutions $\boldsymbol{X}_l$ according to their hypervolume improvement values. The details of the global and local batch selection can be found in Appendix B.2.

The selected solutions $\boldsymbol{X} = \boldsymbol{X}_g \cup \boldsymbol{X}_l$ are then decoded to obtain their molecular representations. These molecules are subsequently evaluated for their true objective vectors $\boldsymbol{Y}$. The pair $\{\boldsymbol{X}, \boldsymbol{Y}\}$ is used to update the trust regions, and the current iteration is over. The next iteration starts with the updated trust regions, allowing the optimization process to iteratively improve and refine the Pareto set. This approach combines both global and local information, leveraging the strength of

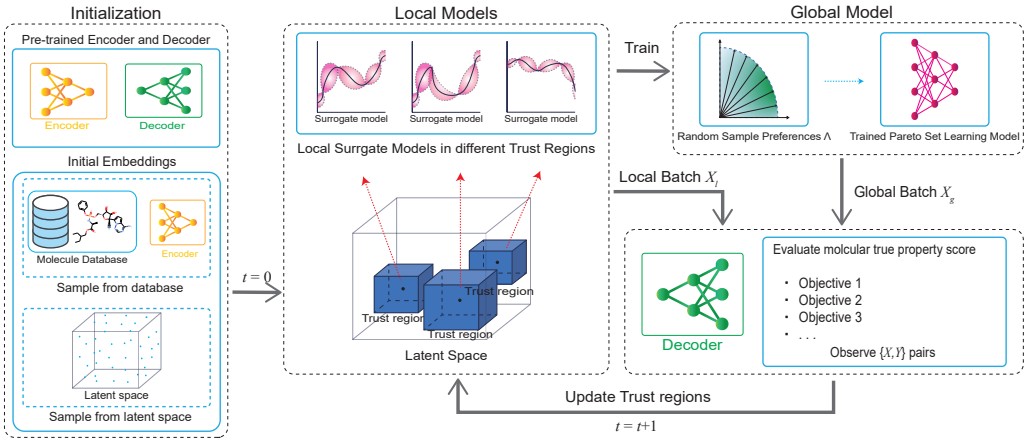

Figure 1: Illustration of the framework of the proposed MLPS

each to enhance the efficiency and effectiveness of the multi-objective molecular design process. We summarize the core components of MLPS in Algorithm 1.

---

**Algorithm 1** The framework of MLPS

---

**Input:** Pre-trained encoder and decoder, initial embeddings and their objective vectors $\{X_{init}, Y_{init}\}$, the number of trust regions $n_{tr}$, the minimum edge length of trust regions $L_{min}$
1: Initialize $n_{tr}$ trust regions $\boldsymbol{T} = \{T_1, T_2, \cdots, T_{n_{tr}}\}$ with $\{X_{init}, Y_{init}\}$;
2: Build a local surrogate model in each trust region;
3: **while** budget not exhausted **do**
4:   Train the global Pareto set learning model $h_{\boldsymbol{\theta}}$ based on the local surrogate models (Alg.3);
5:   Generate solutions from random preferences by $h_{\boldsymbol{\theta}}$ to form the global batch $X_g$ (Alg.5);
6:   Sample solutions in $\boldsymbol{T}$ to form the local batch $X_l$ (Alg.4);
7:   $X \leftarrow X_g \cup X_l$;
8:   Decode $X$ and obtain their true objective vectors $Y$;
9:   **for** $j = 1, 2, \cdots, n_{tr}$ **do**
10:    Update the edge length of $T_j$ based on $\{X, Y\}$;
11:    **if** the edge length of $T_j$ is less than $L_{min}$ **then**
12:     Reinitialize $T_j$ (Alg.2);
13:    **end if**
14:    Update the center of $T_j$ based on $f_{HVC}$ (Subsection 3.2);
15:   **end for**
16:   Update the local surrogate model in each trust region;
17: **end while**
**Output:** global Pareto set learning model $h_{\boldsymbol{\theta}}$

---

## 3.2 LOCAL MODELS

In our MLPS, each trust region is equipped with a local surrogate model, which makes MLPS different from other global Bayesian optimization methods. Specifically, each local surrogate model is a Gaussian process implemented using BoTorch (Balandat et al. (2020)) and GPyTorch (Gardner et al. (2018)). Our approach focuses on maintaining the accuracy of each local surrogate model within its respective trust region. Similar to Turbo (Eriksson et al. (2019)), a trust region is a hypercube region defined within the latent space in this work. There are two key issues for each trust region in our work: the setting of its center and the reinitialization strategy, which we discuss below:

**Center Setting.** In single-objective optimization, existing approaches often place the center of a trust region at the best observed point. However, this approach is no longer suitable for multi-objective optimization since there exists no single best solution. In MLPS, we select the point with the maximum hypervolume contribution as the trust region center. Given a non-dominated solution set within the current trust region, we calculate the hypervolume contribution for each solution and choose the point with the highest hypervolume contribution to be the center of the trust region, while excluding points that have already been selected as the centers of other trust regions. Placing the center of the trust region at the point with the highest hypervolume contribution enhances diversity. This is because more crowded solutions tend to have smaller hypervolume contributions.

**Reinitialization Strategy.** As mentioned in Section 2, an unpromising trust region is penalized by halving the edge length. This reduction in edge length is not unlimited. If the length of an edge becomes smaller than a predefined threshold $L_{min}$, the current trust region is terminated, and a new trust region is generated. For generating a new trust region, we determine the position of the new center and build a new local surrogate model. Let $D_p = (X_p, Y_p)$ be the set of previously reinitialized center points $X_p$ and their corresponding observations $Y_p$. $\hat{f}_r$ is the surrogate model corresponding to the trust region to be reinitialized $T_r$. Then, we build a new local surrogate model to $D_p$: $\hat{f}_{new} \sim P(\hat{f}_r \mid D_p)$. Based on $\hat{f}_{new}$, we identify the center point $x_c$ of the new trust region by maximizing a scalarizing function specified by a random preference. The reinitialization strategy is outlined in Algorithm 2. This approach ensures that trust regions can be reinitialized in promising parts of the overall space.

---

**Algorithm 2** Reinitialization Strategy for Trust Regions

---

**Input:** the trust region to be reinitialized $T_r$, the corresponding local surrogate model $\hat{\boldsymbol{f}}_r$, previously reinitialized center points $\boldsymbol{X}_p$, their observations $\boldsymbol{Y}_p$, initial edge length of trust region $L_{init}$

1: Build a local surrogate model to the reinitialized center points $\boldsymbol{D}_p = (\boldsymbol{X}_p, \boldsymbol{Y}_p)$: $\hat{\boldsymbol{f}}_{new} \sim P(\hat{\boldsymbol{f}}_r \mid \boldsymbol{D}_p)$;
2: Randomly sample a preference $\boldsymbol{\lambda} \sim \boldsymbol{\Lambda} = \{\boldsymbol{\lambda} \in \mathbb{R}^M \mid \sum \lambda_i = 1\}$; ($\cdot_i$ denotes the $i$-th element)
3: $\boldsymbol{x}_c = arg\max_{\boldsymbol{x} \in \boldsymbol{\chi}} s_\lambda[\hat{\boldsymbol{f}}_{new}(\boldsymbol{x})]$, where $s_\lambda[\boldsymbol{y}] = \min(\max(\frac{y_i}{\lambda_i}, 0))$; ($\cdot_i$ denotes the $i$-th element)
4: Obtain the true objective vector of $\boldsymbol{x}_c$, denoted as $\boldsymbol{y}_c$;
5: Reinitialize $T_r$ with the center $\boldsymbol{x}_c$ and the edge length $L_{init}$;
6: $\boldsymbol{X}_p \leftarrow \boldsymbol{X}_p \cup \boldsymbol{x}_c$;
7: $\boldsymbol{Y}_p \leftarrow \boldsymbol{Y}_p \cup \boldsymbol{y}_c$;

**Output:** reinitialized trust region $T_r$

---

## 3.3 GLOBAL PARETO SET LEARNING MODEL

As previously highlighted, the Pareto set of a multi-objective problem may contain an infinite or large number of solutions, each offering a different trade-off among the objectives. In contrast to traditional MMD approaches, MLPS employs a global Pareto set learning model to map a user defined trade-off preference to the corresponding Pareto optimal solution. To achieve this, we employ scalarizing functions to calculate gradients using local surrogate models, and these gradients are utilized to update the global Pareto set learning model. The global Pareto set learning model serves as a bridge connecting different trust regions. It can generate improved global optimal solutions, encouraging local search within trust regions to explore and discover even better solutions. Conversely, The enhanced local solutions found in trust regions can provide guidance to update the global model. This feedback loop ensures that the global model continuously improves its understanding of the Pareto set and adapts to changing preferences.

**Model Formulation.** The function of the global Pareto set learning model is expressed as follows:

$$\boldsymbol{x} = h_{\boldsymbol{\theta}}(\boldsymbol{\lambda}). \tag{2}$$

Here, $\boldsymbol{\lambda}$ represents any valid preference selected from $\boldsymbol{\Lambda} = \{\boldsymbol{\lambda} \in \mathbb{R}^M \mid \sum \lambda_i = 1\}$, with $i = 1, 2, \ldots, M$. $\boldsymbol{x}$ is the corresponding solution to the preference $\boldsymbol{\lambda}$ in the $N$-dimensional latent space. Typically, $N$ is much larger than $M$ in MMD. $h_{\boldsymbol{\theta}}$ denotes a neural network with parameters $\boldsymbol{\theta}$, which we employ to model the complex mapping from a preference to a solution. We choose a multi-layered perceptron with attention mechanisms to enhance the learning process. Specifically, four attention layers are incorporated to capture the subtle distinctions among different preferences effectively. These distinctions play a pivotal role in providing diverse and accurate solutions. To ensure that the model learns these distinctions effectively, we introduce residual connections within the network. These connections help preserve the original preference information throughout the transformation process. By combining preference data with its features, our model can generate corresponding solutions that are more distinguishable and faithful to the preferences.

Once the global Pareto set learning model is adequately trained, it empowers decision-makers to fine-tune their preferences, facilitating exploration across the entire Pareto set. This flexibility allows decision-makers to select solutions that align precisely with their specific needs and trade-off preferences, making our approach adaptable to a wide range of multi-objective molecular design scenarios.

**Model Training.** The training process aims to update the model's parameters $\boldsymbol{\theta}$ such that the generated solutions align with the optimal solutions by minimizing the augmented Tchebycheff scalarization. This can be expressed as:

$$\boldsymbol{x}^* = h_{\boldsymbol{\theta}^*}(\boldsymbol{\lambda}) = \arg\min_{\boldsymbol{x} \in \boldsymbol{\chi}} g_{tch\_aug}(\boldsymbol{x} \mid \boldsymbol{\lambda}). \tag{3}$$

The detaild of the augmented Tchebycheff scalarization $g_{tch\_aug}$ can be found in Appendix B.3. This function establishes a connection between a set of preferences $\boldsymbol{\Lambda} = \{\boldsymbol{\lambda} \in \mathbb{R}^M \mid \sum \lambda_i = 1\}$ and their corresponding solutions within the Pareto set. It guides the global Pareto set learning model to generate solutions that are close to the Pareto front.

To find the optimal parameter $\boldsymbol{\theta}^*$, we propose an efficient algorithm. Since the optimal solution of the augmented Tchebycheff scalarizing function is unknown, we need to optimize all solutions generated by our model with the corresponding augmented Tchebycheff scalarizing functions for all valid preferences:

$$\boldsymbol{\theta}^* = \arg\min_{\boldsymbol{\theta}} E_{\boldsymbol{\theta} \sim \boldsymbol{\Lambda}} g_{tch\_aug}(\boldsymbol{x} = h_{\boldsymbol{\theta}}(\boldsymbol{\lambda}) \mid \boldsymbol{\lambda}) \tag{4}$$

Solving Eq. (4) directly is challenging due to the expectation over an infinite set of preferences. Therefore, we employ a Monte Carlo sampling and gradient descent approach to iteratively update the model with different surrogate models in different trust regions. The update equation is as follows:

$$\boldsymbol{\theta}_{t+1} = \boldsymbol{\theta}_t - \eta \sum_{k=1}^{K} \nabla_{\boldsymbol{\theta}} \hat{g}_{tch\_aug}(\boldsymbol{x} = h_{\boldsymbol{\theta}}(\boldsymbol{\lambda}_k) \mid \boldsymbol{\lambda}_k). \tag{5}$$

Here, $\hat{g}_{tch_aug}(\cdot)$ is the augmented Tchebycheff scalarizing function with objective vectors predicted by local surrogate models (Please refer to Eq. (12) in Appendix B.3). To account for the uncertainty of the surrogate model, we use the lower confidence bound to obtain the surrogate objective vector:

$$\hat{\boldsymbol{f}}(\boldsymbol{x}) = \hat{\boldsymbol{\mu}}(\boldsymbol{x}) - \beta\hat{\boldsymbol{\sigma}}(\boldsymbol{x}), \tag{6}$$

where $\hat{\boldsymbol{\mu}}$ is the mean value, $\hat{\boldsymbol{\sigma}}$ is the variance value, and $\beta$ is a parameter that balances the weight between the mean and variance. In this work, we set $\beta$ to 0.1.

Algorithm 3 describes the training of the global Pareto set learning model. This process iteratively updates the global Pareto set learning model, allowing it to learn the mapping from preferences to corresponding solutions in the Pareto set.

---

**Algorithm 3** Global Pareto Set Learning Model Training

---

**Input:** global Pareto set learning model $h_{\boldsymbol{\theta}}$, the number of iterations for training the global model $T_g$, trust regions $\boldsymbol{T} = \{T_1, T_2, \cdots, T_{n_{tr}}\}$, corresponding local surrogate models $\{\hat{\boldsymbol{f}}_1, \hat{\boldsymbol{f}}_2, \cdots, \hat{\boldsymbol{f}}_{n_{tr}}\}$, the number of sampled preference $n$

1: **for** $t_g = 1, 2, \cdots, T_g$ **do**
2:     Randomly sample $n$ preferences $\{\boldsymbol{\lambda}_1, \boldsymbol{\lambda}_2, \ldots, \boldsymbol{\lambda}_n\}$;
3:     Use the current model $h_{\boldsymbol{\theta}}$ to generate solutions $\boldsymbol{X}_{\boldsymbol{\theta}} = \{\boldsymbol{x} \mid \boldsymbol{x}_i = h_{\boldsymbol{\theta}}(\boldsymbol{\lambda}_i), i = 1, 2, \cdots, n\}$;
4:     Initialize an empty set $loss$;
5:     **for** $i = 1, 2, \cdots, n$ **do**
6:         Find the closest trust region to $\boldsymbol{x}_i$, denoted as $T_j$;
7:         Calculate the scalarizing function $\hat{g}_{tch\_aug}(\boldsymbol{x}_i)$ based on the surrogate model $\hat{\boldsymbol{f}}_j$ (App. B.3);
8:         Calculate $\nabla_{\boldsymbol{\theta}} \hat{g}_{tch\_aug}(\boldsymbol{x}_i)$ and add it to the $loss$ set;
9:     **end for**
10:    Update $\boldsymbol{\theta}$ with $loss$;
11:    $t_g = t_g + 1$;
12: **end for**
**Output:** updated Pareto set learning model $h_{\boldsymbol{\theta}}$

---

# 4 EXPERIMENTS

## 4.1 EXPERIMENT SETUP

**Multi-Objective Molecular Design Scenarios.** In our experiments, we aim to explore the effectiveness of the proposed MLPS in various multi-objective molecular design scenarios. To do so, we consider seven objectives related to molecular properties, following Jin et al. (2020); Xie et al. (2021); Sun et al. (2022); Gao et al. (2022). These objectives consist of two biological objectives, namely GSK3$\beta$ and JNK3, which measure the inhibition scores against glycogen synthase kinase-3$\beta$ and c-Jun N-terminal kinase-3 (associated with Alzheimer's disease) target proteins, respectively. We incorporate two non-biological objectives, druglikeness (QED) (Bickerton et al. (2012)) and synthetic accessibility (SA) (Ertl & Schuffenhauer (2009)), which assess the drug-likeness and synthesizability of molecules, respectively. We also consider three multi-property objectives (MPO) from

the PMO benchmark (Gao et al. (2022)) that are widely acknowledged as challenging: Perindopril-MPO, Ranolazine-MPO, and Zaleplon-MPO. Each of these tasks focuses on designing molecules with high fingerprint similarity to a known target drug, while also differing in specific target ways. To comprehensively evaluate the performance of our approach across different types and numbers of objectives, we consider several objective combinations within our experiments: **Two-objective scenarios**: We investigate scenarios where we optimize for pairs of objectives, which include QED+SA (non-biological objectives) and GSK3$\beta$+JNK3 (biological objectives). These scenarios help us assess the effectiveness of our approach when considering only biological or non-biological objectives. **Three-objective scenarios**: Within this category, we extended our optimization approach to include three objectives, specifically QED+SA combined with each of the remaining objectives. These scenarios aimed to strike a balance between optimizing drug-likeness, synthesizability, and one complex objective simultaneously. **Four-objective scenario**: In this scenario, we simultaneously optimized all four objectives, namely QED+SA+GSK3$\beta$+JNK3. This setup rigorously assessed our approach's capability to handle a diverse set of objectives, covering both biological and non-biological aspects.

**Baseline.** We compare the performance of MLPS against six state-of-the-art methods for molecular design. Here is a brief description of each baseline method: 1) **GA+D** (Nigam et al. (2020)) combines a genetic algorithm with a machine learning-based generative model to enhance the diversity of generated molecules. 2) **JT-VAE**(Jin et al. (2018a)) generates molecules by constructing a tree-structured scaffold over chemical substructures and then assembling them into complete molecules, leveraging variational autoencoders. 3) **GCPN** (You et al. (2018)) employs reinforcement learning to generate molecules atom by atom, utilizing a graph neural network (GNN) as the basis. 4) **RationaleRL** (Jin et al. (2020)) extends molecule rationales into complete molecules using reinforcement learning. 5) **MARS** (Xie et al. (2021)) utilizes Markov sampling to generate molecules using a combination of GNNs and molecule fragments. 6) **MolSearch** (Sun et al. (2022)) employs a Monte Carlo tree search algorithm to discover molecular design moves and generate molecules. 7) **RetMol** (Wang et al. (2023)) is introduces a search-based framework that guides controlled molecule generation using a small set of example molecules, iteratively refining the process to meet desired properties. Please refer to Appendix C.2 for more details of these baseline methods.

**Evaluation Metrics**. We generate 5000 molecules by each method and use **hypervolume (HV)** to compare these methods. HV assesses how well the generated molecules cover the Pareto front in the objective space. It indicates how closely the generated molecules approximate the entire Pareto set and provides insights into their distribution in the objective space across different objectives (see details in Appendix A). To calculate HV, we normalize the range of each objective into $[0, 1]$ and set the reference point be $(0, \ldots, 0)$ in the normalized objective space. Therefore, the range of HV is $[0, 1]$. In addition to HV, we consider several traditional metrics that are commonly used in molecular design. Please refer to Appendix C for details.

## 4.2 RESULTS AND ANALYSIS

Table 1 presents the average HV values over 10 runs obtained by the compared methods. From the results, we can observe that: 1) MLPS consistently stands out as the top-performing method, achieving the highest HV scores in six out of the eight tasks and the second highest in the remaining two. This underscores MLPS's exceptional ability to approximate the entire Pareto set effectively. It excels in delivering diverse and high-quality solutions across a wide range of multi-objective scenarios. 2) While MLPS leads the pack, other methods like MARS, RationaleRL, and RetMol also demonstrate commendable performance in terms of HV scores. These methods exhibit strong capabilities in multi-objective molecular design. However, MLPS outperforms them in most scenarios, highlighting its superiority. 3) Several methods, such as GA+D, JT-VAE, and GCPN, perform acceptably in the two-objective scenarios. However, their performance degrades when faced with problems involving three or four objectives. 4) MolSearch performs relatively better in the four-objective scenario compared to the three-objective scenarios. This behavior might be attributed to the optimization strategy that MolSearch employs: it starts with a specialized molecule set tailored for the four-objective scenario. Note that MolSearch is not applicable to the QED+SA scenario due to its two-stage nature, where the optimization of biological objectives is prioritized in the first stage.

**Ablation Study.** In the ablation study, we aim to assess the contributions of the designed global and local models in MLPS. We compare MLPS with two variants: one that lacks the global model

Table 1: Comparison of different methods on HV

| Method | QED +SA | GSK3$\beta$ +JNK3 | QED+SA +GSK3$\beta$ | QED+SA +JNK3 | QED+SA+ Perindopril-MPO | QED+SA+ Ranolazine-MPO | QED+SA+ Zaleplon-MPO | QED+SA+ GSK3$\beta$+JNK3 |
|---|---|---|---|---|---|---|---|---|
| GA+D | 0.598 | 0.350 | 0.243 | 0.251 | 0.176 | 0.583 | 0.242 | 0.137 |
| JT-VAE | 0.832 | 0.460 | 0.276 | 0.287 | 0.213 | 0.460 | 0.030 | 0.254 |
| GCPN | 0.850 | 0.830 | 0.186 | 0.191 | 0.152 | 0.461 | 0.070 | 0.100 |
| RationaleRL | 0.750 | 0.762 | 0.722 | 0.567 | 0.490 | 0.506 | 0.080 | 0.539 |
| MARS | *0.916* | 0.898 | 0.763 | 0.778 | *0.638* | **0.687** | 0.258 | 0.679 |
| MolSearch | / | 0.723 | 0.183 | 0.217 | 0.461 | 0.398 | 0.090 | 0.571 |
| RetMol | 0.847 | **0.910** | *0.771* | *0.781* | 0.578 | 0.590 | *0.316* | *0.701* |
| MLPS | **0.922** | *0.902* | **0.781** | **0.788** | **0.661** | *0.633* | **0.351** | **0.714** |

and another that lacks the local models. In the context of MLPS without local models, our approach does not employ trust regions for partitioning and local searching within the latent space. Instead, a single Gaussian process surrogate model is utilized to fit all observed solutions across the entire latent space. The results are presented in Fig. 2, which illustrates the Hypervolume (HV) values as a function of the number of solutions that are evaluated by their true objective vectors during the MLPS learning process for these three methods in the context of the QED+SA+GSK3$\beta$+JNK3 task. Notably, the HV values are calculated based on the non-dominated solutions found during the optimization process. We have two key observations from this figure: 1) MLPS generally demonstrates quicker convergence compared to its variants that lack either the global or local models. This indicates that the inclusion of both global and local models accelerates the optimization process, allowing MLPS to approximate the Pareto set more efficiently. 2) Among the variants, MLPS without the global model (but with the local models) consistently outperforms MLPS without the local models (but with the global model). This suggests that leveraging multiple local optimizations is more effective than relying solely on a single global optimization in the high-dimensional latent space. Similar observations to Fig. 2 are observed for the other tasks.

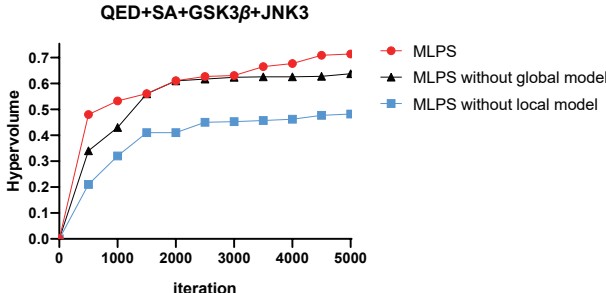

Figure 2: Performance comparison among MLPS and its variants on QED+SA+GSK3$\beta$+JNK3.

We invite readers to examine Appendix C for more detailed experimental analysis and discussions.

## 5 CONCLUSION

This paper proposed an innovative algorithm called Multi-Objective Molecular Design through Learning Latent Pareto Set (MLPS) to address various challenges of multi-objective molecular design (MMD). MLPS leverages a combination of global and local optimization models while learning the mapping from preferences to the Pareto set. This unique approach empowers decision-makers to efficiently explore the Pareto Set. Our extensive experiments across diverse MMD scenarios have consistently demonstrated the superiority of MLPS over state-of-the-art methods. In future studies, we will try to further enhance the scalability and efficiency of MLPS to tackle MMD problems with a higher number of objectives. Moreover, we aspire to extend the application of MLPS to practical domains, such as real-world drug discovery and materials science projects. Since MLPS is designed as an end-to-end optimization framework, combining it with various encoder-decoder architectures and continuous latent representations is also an interesting work.

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

# Appendix

## A    BASIC CONCEPTS IN MULTI-OBJECTIVE OPTIMIZATION

In multi-objective optimization, the goal is to identify a set of solutions that can either maximize or minimize a vector-valued objective function. As mentioned in Section 3, we treat the multi-objective molecular design problem as a maximization problem. Generally, there exists no single solution that can maximize each objective function simultaneously. Therefore, a trade-off among these objectives becomes necessary. These objective vectors are typically compared using Pareto dominance.

**Definition 1 (Pareto Dominance):** Let $x_a, x_b \in \chi$. An objective vector $f(x_a)$ dominates $f(x_b)$, denoted as $f(x_a) \succ f(x_b)$, if $f_i(x_a) \geq f_i(x_b)$ for all $i = 1, 2, \ldots, M$, and there exists a $j \in 1, 2, \ldots, M$ such that $f_j(x_a) > f_j(x_b)$. A solution that is not dominated by any other solution is referred to as a Pareto optimal solution.

**Definition 2 (Pareto Set/Front):** The set comprising all Pareto optimal solutions is termed the Pareto Set, denoted as $\mathbb{U}_{ps} = \{x^* \mid \nexists \ x \succ x^*, x \in \chi\}$, and the corresponding Pareto Front is $f(\mathbb{U}_{ps}) = \{f(x^*) \mid x^* \in \mathbb{U}_{ps}\}$.

**Definition 3 (Hypervolume):** The hypervolume (HV) indicator is employed to quantify how effectively a solution approximates the Pareto Front. As illustrated in Fig 3(a), $r^*$ denotes a pre-defined reference point $((0,0)$ in this case), and $X$ represents a set of solutions in the objective space. The hypervolume indicator, denoted as $f_{HV}(X)$, measures the volume of $S$ (the dark blue region) dominated by $X$:

$$f_{HV}(X) = S = \{r \in \mathbb{R}^M \mid \exists x \in X, f(x) \succ r \succ r^*\}. \tag{7}$$

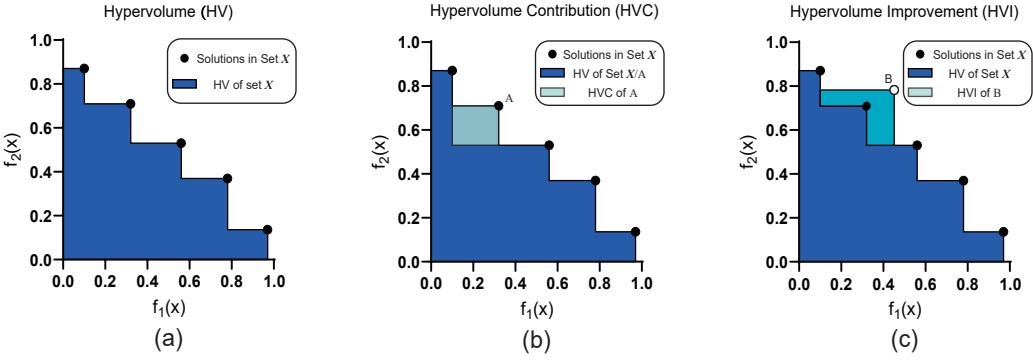

Figure 3: Illustration of the hypervolume indicator.

For a given solution set $X$, we utilize the hypervolume contribution (HVC) to assess the performance of each solution within $X$. A solution's hypervolume contribution represents the difference in hypervolume between $X$ and $X$ when the solution is excluded:

$$f_{HVC}(x \mid X) = f_{HV}(X) - f_{HV}(X/x). \tag{8}$$

As shown in Fig 3(b), the hypervolume contribution of solution A corresponds to the volume of the light blue region.

When introducing a new solution $x_{new}$ into $X$, we employ the hypervolume improvement (HVI) to quantify the difference in hypervolume between $X$ and $X$ with the inclusion of the new solution:

$$f_{HVI}(x_{new} \mid X) = f_{HV}(X \cup x) - f_{HV}(X). \tag{9}$$

As demonstrated in Fig 3(c), the hypervolume improvement of solution B represents the volume of the light blue region.

## B  MORE DETAILS ON THE PROPOSED METHOD

### B.1  MOLECULAR REPRESENTATIONS AND ENCODER-DECODER

There are two common types of molecular representations. One approach represents a molecule as a discrete string of chemical characters, such as SMILES and SELFIES (Krenn et al. (2020)). The other employs graphs or motifs to represent molecules. For guided search molecular optimization, ensuring the validity of the latent space learning through the encoder-decoder is crucial. Invalid solutions can lead to excessive evaluation costs and produce meaningless results.

In the first strategy, SELFIES is a string-based representation of molecules that boasts 100% robustness. In contrast to SMILES, SELFIES can consistently represent valid molecules and encompass all possible molecules. This ensures that every point in the latent space can be successfully decoded as a valid molecule.

The second strategy involves methods that establish a coherent grammar to represent the entire molecular graph or break down the graph into distinct motifs. As atoms and bonds intuitively correspond to nodes and edges, these methods, based on molecular construction, can also represent all molecules while guaranteeing the validity of decoded molecules.

In our MLPS framework, we employ SELFIES to represent molecules, thereby ensuring their validity. Additionally, using string-based representations can help circumvent the computational expenses associated with graph structures. A recent study by Maus et al. (2022) introduced a novel Variational Autoencoder (VAE) architecture based on the SELFIES string representation. This VAE model comprises six transformer encoder and transformer decoder layers, with a latent space dimensionality of 256. We utilize this SELFIES VAE model for all molecular design tasks in our work.

### B.2  BATCH SELECTION

In each iteration, the global and local models generate two sets of solutions, respectively. The best batch of solutions in each set is selected using the hypervolume improvement to update trust regions and guide future generations.

#### B.2.1  LOCAL BATCH SELECTION

For the local batch selection strategy (Algorithm 4), we employ the following approach. The local batch size is denoted as $N_l$, and the local batch selection set, denoted as $\boldsymbol{X}_l$, is initially empty. The subsequent steps are iterated $N_l$ times:
1) In each trust region, we utilize Thompson sampling (Russo et al. (2018)) to generate $n_l$ points, represented as $\boldsymbol{X}_l^j = \{\boldsymbol{x}_1, \boldsymbol{x}_2, ..., \boldsymbol{x}_{n_l}\}$.
2) We predict the objective vectors corresponding to these generated points using the respective surrogate model.
3) Within our MLPS, we maintain a set $\mathbb{U}_{nd}$, which contains non-dominated solutions found thus far. Only solutions evaluated with their true objective vectors are eligible to enter this set. We select the solution with the largest value of $f_{HVI}(\boldsymbol{x}_i \mid \mathbb{U}_{nd})(\boldsymbol{x}_i \in \{\boldsymbol{X}_l^1, \boldsymbol{X}_l^2, \ldots, \boldsymbol{X}_l^{n_{tr}}\})$ to be included in $\boldsymbol{X}_l$. It is worth noting that the predicted objective vectors are utilized in the calculation of $f_{HVI}$.

#### B.2.2  GLOBAL BATCH SELECTION

The global batch selection strategy is outlined in Algorithm 5. This strategy employs a global batch size denoted as $N_g$ and a global sample size of $n_g$. Initially, we sample $n_g$ valid preferences, represented as $\boldsymbol{\Lambda}_g = \{\boldsymbol{\lambda}_1, \boldsymbol{\lambda}_2, \ldots, \boldsymbol{\lambda}_{n_g}\}$. These preferences are used to generate corresponding solutions, forming the set $\boldsymbol{X}_s = \{\boldsymbol{x} \mid \boldsymbol{x}_i = h_{\boldsymbol{\theta}}(\boldsymbol{\lambda}_i), \boldsymbol{\lambda}_i \in \boldsymbol{\Lambda}_g\}$. It is important to note that not all generated points fall within trust regions. Therefore, we utilize the surrogate model associated with the closest trust region to each point to obtain the surrogate objective vector. We then select the best $N_g$ solutions in $\boldsymbol{X}_s$ based on $f_{HVI}(\boldsymbol{x}_i \mid \mathbb{U}_{nd})$ in a sequential greedy manner.

---

**Algorithm 4** Local Batch Selection

---

**Input:** local batch size $N_l$, sample size for each trust region $n_l$, trust regions $\boldsymbol{T} = \{T_1, T_2, \cdots, T_{n_{tr}}\}$, corresponding local surrogate models $\{\hat{\boldsymbol{f}}_1, \hat{\boldsymbol{f}}_2, \ldots, \hat{\boldsymbol{f}}_{n_{tr}}\}$, non-dominated solution set found so far $\mathbb{U}_{nd}$

1:  $\boldsymbol{X}_l = \emptyset$;
2:  **for** $k = 1, 2, \cdots, N_l$ **do**
3:      **for** $j = 1, 2, \ldots, n_{tr}$ **do**
4:          Sample $n_l$ points $\boldsymbol{X}_l^j = \{\boldsymbol{x}_1, \boldsymbol{x}_2, \ldots, \boldsymbol{x}_{n_l}\}$ in $T_j$;
5:          **for** $i = 1, 2, \cdots, n_l$ **do**
6:              Predict the objective vector $\boldsymbol{y}_i = \hat{\boldsymbol{f}}_j(\boldsymbol{x}_i)$;
7:          **end for**
8:      **end for**
9:      $\boldsymbol{x}_{\max} = \arg\max_{\boldsymbol{x}_i \in \{\boldsymbol{X}_l^1, \boldsymbol{X}_l^2, \ldots, \boldsymbol{X}_l^{n_{tr}}\}} f_{HVI}(\boldsymbol{x}_i \mid \mathbb{U}_{nd})$ ($\boldsymbol{y}_i$ is used to calculate hypervolume);
10:     $\boldsymbol{X}_l = \boldsymbol{X}_l \cup \boldsymbol{x}_{\max}$;
11: **end for**

**Output:** local batch solution set $\boldsymbol{X}_l$

---

**Algorithm 5** Global Batch Selection

---

**Input:** Pareto set learning model $h_{\boldsymbol{\theta}}$, global batch size $N_g$, global sample size $n_g$, trust regions $\boldsymbol{T} = \{T_1, T_2, \cdots, T_{n_{tr}}\}$, corresponding local surrogate models $\{\hat{\boldsymbol{f}}_1, \hat{\boldsymbol{f}}_2, \ldots, \hat{\boldsymbol{f}}_{n_{tr}}\}$, non-dominated solution set found so far $\mathbb{U}_{nd}$

1:  Randomly sample $n_g$ preferences $\boldsymbol{\Lambda}_g = \{\boldsymbol{\lambda}_1, \boldsymbol{\lambda}_2, \ldots, \boldsymbol{\lambda}_{n_g}\}$;
2:  $\boldsymbol{X}_s = \{\boldsymbol{x} \mid \boldsymbol{x}_i = h_{\boldsymbol{\theta}}(\boldsymbol{\lambda}_i), \boldsymbol{\lambda}_i \in \boldsymbol{\Lambda}_g\}$;
3:  **for** $i = 1, 2, \ldots, n_g$ **do**
4:      Find the closest trust region to $\boldsymbol{x}_i$, denoted as $T_j$;
5:      Predict the objective vector $\boldsymbol{y}_i = \hat{\boldsymbol{f}}_j(\boldsymbol{x}_i)$;
6:  **end for**
7:  $\boldsymbol{X}_g = \emptyset$;
8:  **for** $i = 1, 2, \ldots, N_g$ **do**
9:      $\boldsymbol{x}_{\max} = \arg\max_{\boldsymbol{x}_i \in \boldsymbol{X}_s} f_{HVI}(\boldsymbol{x}_i \mid \mathbb{U}_{nd})$ ($\boldsymbol{y}_i$ is used to calculate hypervolume);
10:     $\boldsymbol{X}_g \leftarrow \boldsymbol{X}_g \cup \boldsymbol{x}_{\max}$;
11:     $\boldsymbol{X}_s \leftarrow \boldsymbol{X}_s / \boldsymbol{x}_{\max}$;
12: **end for**

**Output:** global batch solution set $\boldsymbol{X}_g$

---

### B.3 Scalarization

In MLPS, we employ the scalarization method to train the global Pareto set learning model, establishing a connection between preferences $\boldsymbol{\Lambda} = \{\boldsymbol{\lambda} \in \mathbb{R}^M \mid \sum \lambda_i = 1\}$ and their corresponding solutions within the Pareto set. The scalarization method guides the global Pareto set learning model to generate solutions that are close to the Pareto front. In MLPS, we utilize the weighted Tchebycheff approach:

$$g_{tch}(\boldsymbol{x} \mid \boldsymbol{\lambda}) = \max_{1 \leq i \leq M} \{\lambda_i((z_i^* + \epsilon) - f_i(\boldsymbol{x}))\}, \tag{10}$$

Here, $\boldsymbol{Z}^* = [z_1^*, z_2^*, \cdots, z_M^*]$ represents the ideal vector for the objective vector $\boldsymbol{f}(\boldsymbol{x})$, which serves as the upper bound for maximization problems. $\epsilon$ is a small positive scalar. $u_i = z_i^* + \epsilon$ is called unachievable utopia value for the $i$-th objective $f_i(\boldsymbol{x})$. In this work, we set $\epsilon = 0.1|\boldsymbol{Z}^*|$.

In common form of Tchebycheff scalarization approach, the solutions found are in a weakly Pareto optimal set actually. There are some solutions in this set which is not desirable for decision-makers. Thus, we choose an augmented Tchebycheff approach to avoid the issue. The augmented approach is as following:

$$g_{tch\_aug}(\boldsymbol{x} \mid \boldsymbol{\lambda}) = \max_{1 \leq i \leq M} \{\lambda_i((z_i^* + \epsilon) - f_i(\boldsymbol{x}))\} + \rho \sum_{i=1}^{M} \lambda_i f_i(\boldsymbol{x}), \tag{11}$$

where $\rho$ is a small positive scalar. In this work, we set $\rho = 0.01$, which is consistent with the setting in ParEGO (Knowles (2006)). With the augmentation term, the weak Pareto optimal solutions are assigned smaller scalarized values than the strong Pareto optimal solutions in Equation 11. This modification ensures that the scalarization approach selects more desirable solutions for decision-makers.

When we use predicted objective vectors by local surrogate models, the augmented Tchebycheff scalarizing function $\hat{g}_{tch_aug}(\cdot)$ is defined as:

$$\hat{g}_{tch\_aug}(\boldsymbol{x} \mid \boldsymbol{\lambda}) = \max_{1 \leq i \leq M} \{\lambda_i((z_i^* + \epsilon) - \hat{f}_i(\boldsymbol{x}))\} + \rho \sum_{i=1}^{M} \lambda_i \hat{f}_i(\boldsymbol{x}), \tag{12}$$

where $\hat{f}_i(\cdot)$ is the surrogate function for $i$-th objective.

## C More Details on Experiments

### C.1 Implementation Details of MLPS

To ensure the robustness of the results, we conduct each algorithm independently 10 times with different random seeds. For the compared methods, we followed the parameter configurations outlined in their respective original studies to ensure well-tuned performance.

As for our MLPS, we allocated a total budget of 5000 expensive evaluations. This budget represents the maximum number of molecules that can be evaluated with true objective vectors. The initial embeddings consist of 500 molecules, and the approach to obtain these initial embeddings varies depending on the specific task. For tasks that involve at least one biological objective, we sample positive molecules from the database to serve as the initial embeddings. On the other hand, for tasks that solely consist of non-biological objectives, we directly sample results from the latent space to form the initial embeddings.

The global Pareto set learning model is trained for a total of 2000 iterations ($T_g$), with each training iteration involving 1000 sampled preferences ($n$). During the global batch selection phase, 1000 preferences and their corresponding solutions are randomly sampled using the trained global model ($n_g$), and subsequently, 20 solutions are selected ($N_g$). The number of trust regions is set to 5 ($n_{tr}$). In the case of local batch selection, 4096 solutions are sampled from each trust region ($n_l$), resulting in a total of 50 solutions being selected across all trust regions ($N_l$).

### C.2 More Details of Baseline Methods

Our approach to utilizing the baseline methods, including GA+D, JT-VAE, GCPN, RationaleRL, MARS, MolSearch, and RetMol, is detailed as follows:

(1) The final 5000 molecules for comparison: For generative models like GCPN, RationaleRL, MARS, and our MLPS, we obtained 5000 molecules through random sampling from each trained model. GA+D, a genetic algorithm, provided its final population of 5000 for comparison. JT-VAE uses Bayesian optimization to search iteratively. The best 5000 molecules during the search process are collected. MolSearch and RetMol maintain an archive of 5000 molecules, where molecules meeting predefined thresholds for each objective are retained.

(2) Handling multi-objective scenarios: Methods such as GA+D, JT-VAE, GCPN, RationaleRL, and MARS employ scalarization based on specific preference vectors, guided by domain knowledge, to address multi-objective scenarios. These specific preference vectors were set in accordance with the original studies' recommendations. MolSearch employs Pareto dominance, while RetMol uses predefined thresholds for each objective to identify desirable molecules.

(3) Use of multi-objective Bayesian optimization: Among the compared methods, JT-VAE is the only one, aside from MLPS, that utilizes Bayesian optimization. JT-VAE employs single-objective Bayesian optimization to optimize its scalarization function, whereas MLPS uses multi-objective Bayesian optimization in the latent space.

### C.3 COMPLEXITY AND RUNNING TIME ANALYSIS

Our MLPS, involving the division of the latent space and the use of local models for exploration, significantly reduces computational costs, especially in high-dimensional and large data scenarios. Each local model is responsible for a specific trust region, fitting only the data points within that region, which effectively manages computational complexity.

Considering a total of $N_{total}$ data points distributed across $n_{tr}$ trust regions, with $\eta$ representing the average overlap of data points among trust regions, each region handles approximately $\eta N_{total}/n_{tr}$ points. Given the cubic time complexity ($O(N_{total}^3)$) for model fitting, the total complexity for all trust regions is $O(n_{tr}(\eta N_{total}/n_{tr})^3) = O(\eta^3 N_{total}^3/n_{tr}^2)$, yielding an asymptotic speedup of $O(n_{tr}^2/\eta^3)$. As trust regions shrink over time, $\eta$ decreases, further enhancing efficiency.

Regarding scalability with increasing objectives, our method remains robust. The surrogate model's complexity and the number of oracles are the primary factors affecting time. Our use of independent Gaussian processes, with a linear increase in a cubic time complexity for each additional objective, ensures manageable scalability.

We evaluated MLPS's running time on various tasks. The results are shown in the following table.

Table 2: Running time of MLPS on various tasks

| Task | Running Time |
|---|---|
| QED+SA | 2.83 hours |
| JNK3 + GSK3$\beta$ | 2.97 hours |
| JNK3+QED+SA | 4.21 hours |
| GSK3$\beta$+QED+SA | 4.18 hours |
| JNK3+GSK3$\beta$+QED+SA | 6.6 hours |

For comparison, the JNK3+GSK3$\beta$+QED+SA task's running times for other methods are shown in the following table.

Table 3: Running time of different methods on JNK3+GSK3$\beta$+QED+SA

| Method | Running Time |
|---|---|
| RationaleRL | 5.8 hours |
| GA+D | 4 hours |
| MARS | 10 hours |
| Molsearch | 7.2 hours |
| MLPS | 6.6 hours |

These times, measured in hours, are relatively negligible compared to the months or years typically required in the conventional drug discovery process.

## C.4 COMPARISON USING TRADITIONAL EVALUATION METRICS

We consider several traditional metrics that are commonly used in molecular design (Jin et al. (2020); Xie et al. (2021); Sun et al. (2022)), even though they were not explicitly designed for multi-objective molecular design:

**Success rate (SR)** represents the percentage of generated molecules that meet the specified criteria for chosen objectives. The criteria include QED$\geq$0.6, SA$\geq$0.67, GSK3$\beta\geq$0.5, and JNK3$\geq$0.5. SR indicates how well the generated molecules align with the desired properties.

**Novelty (Nov)** quantifies the percentage of generated molecules that are dissimilar (similarity less than 0.4) to the nearest neighbor in the training set (Olivecrona et al. (2017)). This metric assesses the uniqueness of the generated molecules.

**Diversity (Div)** quantifies the dissimilarity between pairs of generated molecules. It is calculated using pairwise Tanimoto similarity over Morgan fingerprints. Div reflects the structural diversity of the generated molecules.

**Product Metric (PM)** is a composite metric that combines SR, Nov, and Div into a single value. It provides a holistic evaluation of the generated molecules, considering their effectiveness, diversity, and novelty simultaneously.

**Results.** Tables 4, 5, and 6 give the average SR, Nov, Div, and PM values over 10 runs obtained by the compared methods. These results reveal several key insights into the performance of MLPS compared to the baseline methods:

1) Across all tasks, MLPS consistently outperforms all baseline methods in terms of the PM. This highlights MLPS's ability to deliver a well-rounded set of molecules that excels in effectiveness, diversity, and novelty.

2) MLPS does not achieve a 100% SR value in these cases, as it aims to provide the entire Pareto set, which may include molecules that do not meet specific criteria (e.g., QED$\geq$0.6, SA$\geq$0.67, GSK3$\beta\geq$0.5, and JNK3$\geq$0.5). SR alone may not fully capture the quality of the generated molecules, and other metrics should be considered in conjunction.

3) MLPS does not consistently attain the highest Nov and Div values. This can be attributed to the nature of these metrics. High diversity (higher Div) may encompass molecules scattered across the entire chemical space, while an optimal set may concentrate on a specific region of interest. Additionally, molecules in the database (i.e., the training set) may already be close to this optimal region. Thus, evaluating Nov and Div should involve considering other metrics to gain a comprehensive understanding of the generated molecule set.

Table 4: Comparison of different methods on SR, Nov, Div, and PM on the two-objective scenarios

| Objective | QED+SA | | | | GSK3$\beta$+JNK3 | | | |
|---|---|---|---|---|---|---|---|---|
| Method | SR | Nov | Div | PM | SR | Nov | Div | PM |
| GA+D | 0.910 | 1.000 | 0.470 | 0.428 | 0.850 | 1.000 | 0.420 | 0.360 |
| JT-VAE | 0.034 | 0.078 | 0.893 | 0.002 | 0.033 | 0.079 | 0.883 | 0.002 |
| GCPN | 0.036 | 0.089 | 0.887 | 0.003 | 0.035 | 0.080 | 0.874 | 0.002 |
| RationaleRL | 0.876 | 0.971 | 0.842 | 0.716 | 0.842 | 0.981 | 0.831 | 0.686 |
| MARS | 0.996 | 0.793 | 0.681 | 0.538 | 0.995 | 0.753 | 0.691 | 0.518 |
| MolSearch | / | / | / | / | 1.000 | 0.787 | 0.826 | 0.650 |
| RetMol | 0.857 | 0.748 | 0.721 | 0.462 | 0.847 | 0.736 | 0.700 | 0.436 |
| MLPS | 0.876 | 0.931 | 0.967 | **0.789** | 0.834 | 0.914 | 0.963 | **0.734** |

Table 5: Comparison of different methods on SR, Nov, Div, and PM on the three-objective scenarios

| Objective | QED+SA+GSK3$\beta$ | | | | QED+SA+JNK3 | | | |
|---|---|---|---|---|---|---|---|---|
| Method | SR | Nov | Div | PM | SR | Nov | Div | PM |
| GA+D | 0.890 | 1.000 | 0.680 | 0.610 | 0.860 | 1.000 | 0.500 | 0.430 |
| JT-VAE | 0.096 | 0.958 | 0.680 | 0.063 | 0.218 | 1.000 | 0.600 | 0.131 |
| GCPN | 0.000 | 0.000 | 0.000 | 0.000 | 0.000 | 0.000 | 0.000 | 0.000 |
| RationaleRL | 0.891 | 0.341 | 0.891 | 0.270 | 0.787 | 0.190 | 0.874 | 0.131 |
| MARS | 0.995 | 0.950 | 0.719 | 0.680 | 0.913 | 0.948 | 0.779 | 0.674 |
| MolSearch | 1.000 | 0.821 | 0.856 | 0.702 | 1.000 | 0.783 | 0.831 | 0.651 |
| RetMol | 0.913 | 0.813 | 0.702 | 0.521 | 0.951 | 0.842 | 0.722 | 0.578 |
| MLPS | 0.875 | 0.960 | 0.873 | **0.737** | 0.882 | 0.951 | 0.861 | **0.722** |

Table 6: Comparison of different methods on SR, Nov, Div, and PM on the four-objective scenario

| Objective | QED+SA+GSK3$\beta$+JNK3 | | | |
|---|---|---|---|---|
| Method | SR | Nov | Div | PM |
| GA+D | 0.860 | 1.000 | 0.360 | 0.310 |
| JT-VAE | 0.054 | 1.000 | 0.277 | 0.015 |
| GCPN | 0.000 | 0.000 | 0.000 | 0.000 |
| RationaleRL | 0.750 | 0.555 | 0.706 | 0.294 |
| MARS | 0.923 | 0.824 | 0.719 | 0.547 |
| MolSearch | 1.000 | 0.818 | 0.811 | 0.664 |
| RetMol | 0.969 | 0.862 | 0.732 | 0.611 |
| MLPS | 0.861 | 0.912 | 0.852 | **0.669** |

## C.5 COMPARISON USING INDIVIDUAL PERFORMANCE METRICS

Tables 7 and 8 show the results of individual performance metrics, such as average property score and average property score of top molecules on the QED+SA and JNK3+GSK3$\beta$ tasks.

Table 7: The results of individual performance metrics on QED+SA

| | Top-10 Avg QED | Top-10 Avg SA | Top-50 Avg QED | Top-50 Avg SA | Avg QED | Avg SA | Avg Rank |
|---|---|---|---|---|---|---|---|
| GA+D | 0.7628 | 0.9385 | 0.6874 | 0.7917 | 0.6567 | 0.7712 | 6 |
| JT-VAE | 0.9192 | 0.8699 | 0.8893 | 0.8336 | 0.7496 | 0.7206 | 3 |
| GCPN | 0.9114 | 0.8958 | 0.8828 | 0.8519 | 0.6098 | 0.5954 | 5 |
| RationaleRL | 0.8224 | 0.8995 | 0.7697 | 0.8861 | 0.6636 | 0.7488 | 4 |
| MARS | 0.9369 | 0.9998 | 0.9076 | 0.9799 | 0.5997 | 0.8058 | 2 |
| MLPS | 0.9424 | 0.9924 | 0.9307 | 0.9811 | 0.6974 | 0.8159 | 1 |

Table 8: The results of individual performance metrics on JNK3+GSK3$\beta$

| | Top-10 Avg JNK3 | Top-10 Avg GSK3$\beta$ | Top-50 Avg JNK3 | Top-50 Avg GSK3$\beta$ | Avg JNK3 | Avg GSK3$\beta$ | Avg Rank |
|---|---|---|---|---|---|---|---|
| GA+D | 0.8453 | 0.8682 | 0.8072 | 0.8572 | 0.5671 | 0.6182 | 6 |
| JT-VAE | 0.8483 | 0.8782 | 0.8058 | 0.8782 | 0.5793 | 0.5988 | 5 |
| GCPN | 0.8584 | 0.8947 | 0.8061 | 0.8609 | 0.5887 | 0.5665 | 4 |
| RationaleRL | 0.7670 | 0.9620 | 0.7408 | 0.9448 | 0.5883 | 0.7455 | 3 |
| MARS | 0.8910 | 0.9310 | 0.8118 | 0.8888 | 0.5919 | 0.6630 | 2 |
| MLPS | 0.9015 | 0.9573 | 0.8282 | 0.9582 | 0.6083 | 0.7382 | 1 |

Our MLPS model demonstrated superior performance in most metrics, achieving the highest average ranking.

## C.6 VISUALIZATION IN THE OBJECTIVE SPACE

In Figures 4 and 5, the non-dominated solution set in the objective space obtained by each compared method on QED+SA and QED+SA+GSK3$\beta$+JNK3 are visualized, respectively. Figure 5 employs a polar coordinate system (He & Yen (2015)) to represent the distribution in the four-dimensional

objective space. This system is based on uniformly distributed direction vectors defined in the four-dimensional objective space. Solutions are then mapped to the nearest vectors, and the resulting polar graph reveals their distribution. Points in this polar graph that are more diverse and closer to the edge indicate a better solution set. It's evident from these figures that the solution set acquired through our MLPS is generally more diverse and well-converged compared to those obtained by the other methods.

However, it is worth noting that even MLPS obtains a limited number of non-domiated solutions on these tasks. We hypothesize two primary reasons for this observation:

(1) Given the discrete nature of multi-objective molecular design tasks, the actual number of true Pareto optimal solutions is inherently limited. Consequently, even an accurate method for identifying the true Pareto front will yield only a finite set of solutions. This limitation is particularly pronounced in real-world problems like multi-objective molecular design, where the true Pareto front is often unknown. Despite this, our MLPS consistently demonstrates superior performance in approximating the Pareto front, as evidenced by both quantitative metrics such as the hypervolume and visualization results.

(2) The global model within our MLPS framework might not be optimally trained or architecturally perfect. We observed that increasing the evaluation budget (beyond the 5000 used in our current experiments) enhances the HV values achieved by MLPS, suggesting potential for further improvement. Thus, enhancing MLPS's performance through fine-tuning and architectural improvements will be a focus of our future work.

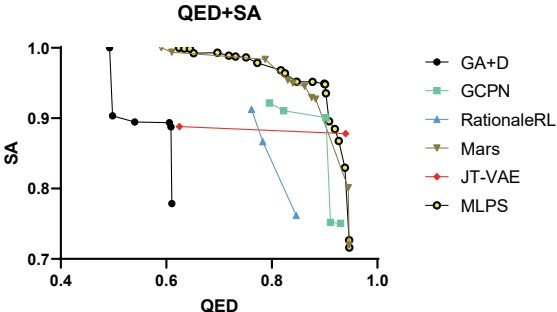

Figure 4: The non-dominated solution set in the objective space obtained by each compared method on QED+SA.

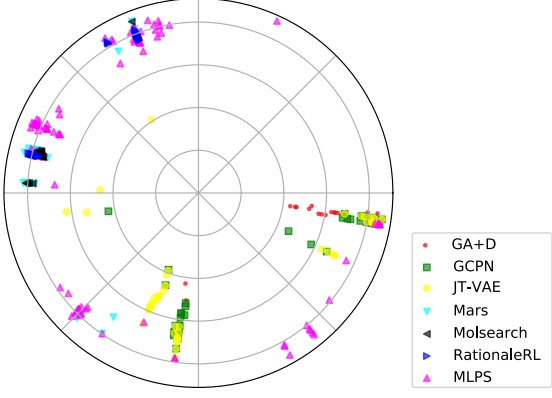

Figure 5: The non-dominated solution set in the objective space (using a polar coordinate system) obtained by each compared method on QED+SA+GSK3$\beta$+JNK3.

## C.7    VISUALIZATION IN THE SOLUTION SPACE

In Figure 6, we utilize t-SNE (Van der Maaten & Hinton (2008)) to visually represent the distribution of generated molecules on the QED+SA+GSK3$\beta$+JNK3 task. We employ the ECFP6 fingerprints, following the methodology from Li et al. (2018). Several notable observations can be made from Figure 6: The generated molecules from GA+D tend to cluster together in two closely located groups, indicating a relatively low diversity. GCPN, JT-VAE, and MolSearch mainly concentrate on a compact region. Molecules generated by RationaleRL and RetMol exhibit a tendency to form distinct groups. Notably, the groups produced by RationaleRL showcase higher diversity compared to those generated by RetMol. Both MLPS and MARS produce molecules with a well-dispersed distribution in the space, suggesting their ability to generate molecules with high novelty and diversity. Notably, the distribution of MLPS covers a larger area compared to that of MARS.

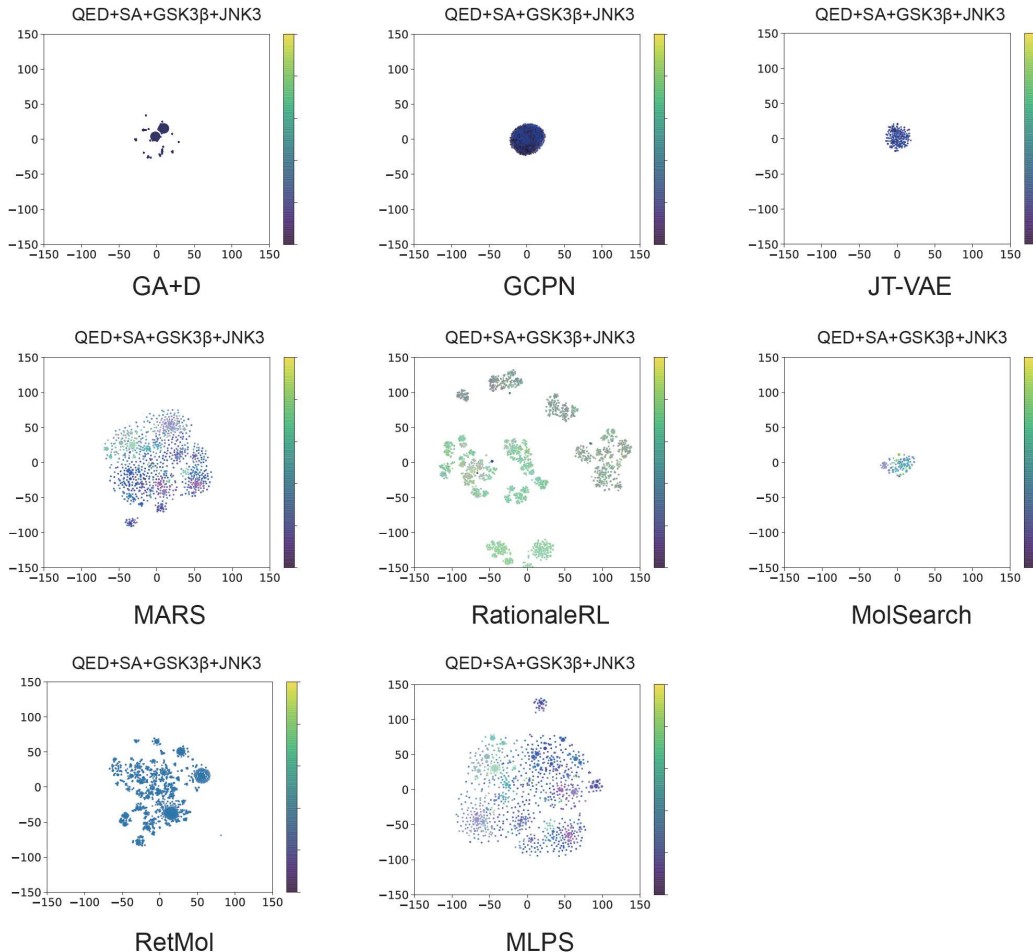

Figure 6: t-SNE visualization of generated molecules on QED+SA+GSK3$\beta$+JNK3.

## C.8    COMPARISON WITH PSL

PSL (Lin et al. (2022)) operates in a continuous space and is applicable to our molecular optimization tasks. We integrated PSL with the same encoder-decoder framework as our MLPS and compared their performance on various tasks. The results are shown in Table 9, indicating that MLPS outperforms PSL in terms of hypervolume. These findings underscore MLPS's superior performance over PSL across the evaluated tasks.

Table 9: The results of HV obtained by MLPS and PSL on various tasks

| Tasks | PSL | MLPS |
|---|---|---|
| QED+SA | 0.763 | 0.922 |
| QED+SA+JNK3+GSK3$\beta$ | 0.432 | 0.714 |
| QED+SA+Ranolazine-MPO | 0.579 | 0.701 |

## C.9 COMPARISON WITH NAS METHODS

Our work in molecular optimization and existing approaches in neural architecture search (NAS) (Li et al. (2020); Lu et al. (2020); Shi et al. (2020); Dudziak et al. (2020)) share similar ideas. Both domains share the common strategy of mapping discrete data to a continuous space for optimization, followed by decoding to obtain optimized solutions. We have adapted the methods from references Li et al. (2020); Lu et al. (2020); Shi et al. (2020) for molecular optimization by substituting their encoder-decoder components. This modification aligns these NAS approaches with the structural requirements of molecular optimization.

To demonstrate the efficacy of our MLPS in comparison to adapted NAS methods, we conducted experiments on two tasks, QED+SA and JNK3+GSK3$\beta$. The hypervolume results are shown in Table 10. These results clearly demonstrate MLPS's superior performance compared to the adapted NAS methods.

Table 10: The results of HV obtained by MLPS and adapted NAS methods on various tasks

| Methods | QED+SA | JNK3+GSK3$\beta$ |
|---|---|---|
| NGAE (Li et al. (2020)) | 0.302 | 0.182 |
| MSuNAS (Lu et al. (2020)) | 0.744 | 0.353 |
| BONAS (Shi et al. (2020)) | 0.631 | 0.367 |
| MLPS | 0.922 | 0.902 |

## C.10 SENSITIVITY ANALYSIS TO HYPERPARAMETERS

This subsection gives the experimental results that systematically evaluate MLPS's sensitivity to hyperparameter variations, accompanied by a detailed discussion of their impacts. We chose the QED+SA task as a representative case study, with similar trends observed in other tasks. Our hyperparameter settings are summarized in the table below. While investigating specific hyperparameters, all others were held constant as per these values:

| Hyperparameter | Value |
|---|---|
| The number of trust regions ($n_{tr}$) | 5 |
| The maximum edge length of trust regions ($L_{\max}$) | 1.6 |
| The minimum edge length of trust regions ($L_{\min}$) | 0.01 |
| The number of preferences sampled for training the global model($n$) | 1000 |
| The number of iterations for training the global model ($T_g$) | 2000 |

(1) The effect of the number of trust regions:

The number of trust regions crucially dictates the latent space's segmentation. A deficient count leads to inadequate exploration and potential high-quality solution omissions. Excessively many regions cause overlapping and redundant searches. We varied the trust regions from 1 to 15, assessing the hypervolume (HV) metric. Results are as follows:

Table 11: Results of HV with different number of trust regions

| The number of trust regions | HV |
|---|---|
| 1 | 0.793 |
| 3 | 0.881 |
| 5 | 0.922 |
| 10 | 0.922 |
| 15 | 0.922 |

The data indicates that too few regions (1 or 3) lead to insufficient exploration and lower HV values, suggesting inadequate global model training. However, a moderate number (5 and above) maintains consistent HV values (0.922), suggesting efficient exploration without further gains beyond this point.

(2) The effect of the maximum and minimum edge lengths of trust regions:

These parameters constrain the trust region size. Excessively large maximum lengths may cause overlapping, while too small lengths limit exploration. Conversely, large minimum lengths can trigger frequent reinitialization, missing potential solutions, whereas too small minimum lengths lead to over-exploitation and low efficiency.

We experimented with various maximum and minimum lengths, seeking a robust setting. Initial edge length was set at 0.8. Table 12 shows the results.

Table 12: Results of HV with different settings of edge length

| Max edge length | Min edge length | Initial edge length | HV |
|---|---|---|---|
| 1.6 | 0.01 | 0.8 | 0.922 |
| 1.6 | 0.2 | 0.8 | 0.879 |
| 1.6 | 0.4 | 0.8 | 0.841 |
| 3.2 | 0.01 | 0.8 | 0.922 |

We can observe that optimal results were achieved with max and min lengths of 1.6 and 0.01. Increasing the max length (e.g,. 3.2) didn't enhance HV, while larger min lengths (e.g,. 0.2 or 0.4) decreased HV, indicating low search efficiency.

(3) The effect of the number of iterations for training the global model

More iterations generally enhance the global model, yielding a more accurate Pareto set. We varied the iterations and observed the HV:

Table 13: Results of HV with different number of iterations for training the global model

| The number of iterations for training the global model | HV |
|---|---|
| 500 | 0.832 |
| 1000 | 0.872 |
| 2000 | 0.922 |
| 5000 | 0.922 |

The HV increases with more iterations, peaking at 2000. Further increase (e.g., to 5000) doesn't change the HV value, indicating an optimal training length.

(4) The effect of the number of preferences sampled for training the global model

In each round of training the global model, we sample a batch of preferences (i.e., direction vectors in the objective space), and then use these vectors to generate solutions and calculate gradients. Sampling sufficient preferences is crucial for adequately covering the Pareto front. We experimented with varying numbers of sampled preferences:

Table 14: Results of HV with different number of preferences sampled for training the global model

| The number of preferences sampled for training the global model | HV |
|---|---|
| 10 | 0.641 |
| 100 | 0.793 |
| 500 | 0.871 |
| 1000 | 0.922 |
| 2000 | 0.922 |

Increasing the number of preferences enhances HV results. However, beyond 1000, HV remains constant, suggesting an optimal sampling rate.

### C.11 MORE DISCUSSIONS AND ANALYSIS ON TRUST REGIONS

In this subsection, we give a more detailed discussion and empirical analysis on trust region management. We address two crucial aspects:

(1) Frequency of Trust Region Intersection and Collapse

The right balance in the number of trust regions is vital to avoid excessive intersection or collapse. A few intersecting trust regions can be beneficial, creating an informative interplay that might lead to better solutions than isolated exploration. To investigate the degree of overlap, we performed experiments on the QED+SA task, focusing on the degree of overlap between trust regions. We measured the distance in each dimension between the centers of trust regions, calculating the average minimum distance among all dimensions ($d_{avg-\min}$) and the average length of trust region edges ($d_{avg-edge}$). When the values of $d_{avg-\min}$ and $d_{avg-edge}$ are close, trust regions may overlap slightly. However, as the value of $d_{avg-edge}$ becomes larger than $d_{avg-\min}$, trust regions are more likely to exhibit increased overlap. Our settings were as follows: 5 trust regions, a budget of 5000, and a batch size of 70 (20 for global and 50 for local), resulting in 71 iterations. The results are:

Table 15: $d_{avg-\min}$ and $d_{avg-edge}$ during the search process

| Iterations | $d_{avg-\min}$ | $d_{avg-edge}$ |
|---|---|---|
| 0 | 0.98 | 0.8 |
| 15 | 0.80 | 0.8 |
| 30 | 0.38 | 0.4 |
| 45 | 0.32 | 0.32 |
| 60 | 0.20 | 0.22 |
| 71 | 0.19 | 0.145 |

These findings indicate slight overlap during most of the search process, suggesting effective communication and exploration among trust regions. Towards the end, the focus shifts to exploitation, as indicated by the smaller $d_{avg-edge}$ compared to $d_{avg-\min}$

(2) Frequency of Trust Region Reinitialization

Trust region reinitialization is critical for ensuring sufficient exploration. Our goal is to avoid frequent reinitializations, which could indicate a lack of thorough exploitation within each region. The following table illustrates the reinitialization frequency in relation to the budget on the QED+SA task:

Table 16: Reinitialization frequency in relation to the budget

| Budget | HV | Number of reinitializations |
|---|---|---|
| 2500 | 0.901 | 1 |
| 5000 | 0.922 | 2 |
| 8000 | 0.926 | 5 |
| 15000 | 0.928 | 7 |

This moderate frequency of reinitialization under our settings indicates a balanced approach between exploration and exploitation.

## C.12    CHOICE OF SCALARIZATION

We have two reasons for selecting Tchebycheff scalarization in our MLPS framework.

(1) Tchebycheff scalarization has a distinctive advantage in its tendency to explore all Pareto optimal solutions. By focusing on the worst-case deviations for each objective, it inherently promotes a thorough understanding of the trade-offs involved among objectives. In contrast, linear scalarization theoretically struggles to identify all Pareto optimal solutions due to its inherent limitations in handling certain trade-offs.

(2) Our empirical results further substantiate the effectiveness of Tchebycheff scalarization in multi-objective molecular design tasks. We conducted comparative experiments between MLPS implementations using Tchebycheff and linear scalarizations across various tasks. The results, as presented below, clearly demonstrate the superior performance of Tchebycheff scalarization in terms of hypervolume (HV):

Table 17: Results of HV obtained by MLPS with different scalarizations

|  | QED+SA | JNK3 + GSK3$\beta$ | QED+SA+JNK3 + GSK3$\beta$ |
|---|---|---|---|
| MLPS (Tchebycheff scalarization) | 0.922 | 0.902 | 0.714 |
| MLPS (linear scalarization) | 0.876 | 0.782 | 0.581 |

These findings affirm our choice of Tchebycheff scalarization for its robustness, comprehensive exploration capabilities, and empirically validated performance.

## C.13    VISUALIZATION OF GENERATED MOLECULES

Figure 7 showcases the non-dominated molecules generated by MLPS for the GSK3$\beta$+JNK3+QED+SA task. Beneath each molecular graph, we have provided the corresponding scores for GSK3$\beta$, JNK3, QED, and SA. This visualization aims to offer a clear representation of the molecular structures generated by our model, along with their respective multi-objective optimization scores.

Figure 7: Non-dominated molecules generated by MLPS on QED+SA+GSK3$\beta$+JNK3. (next four pages)

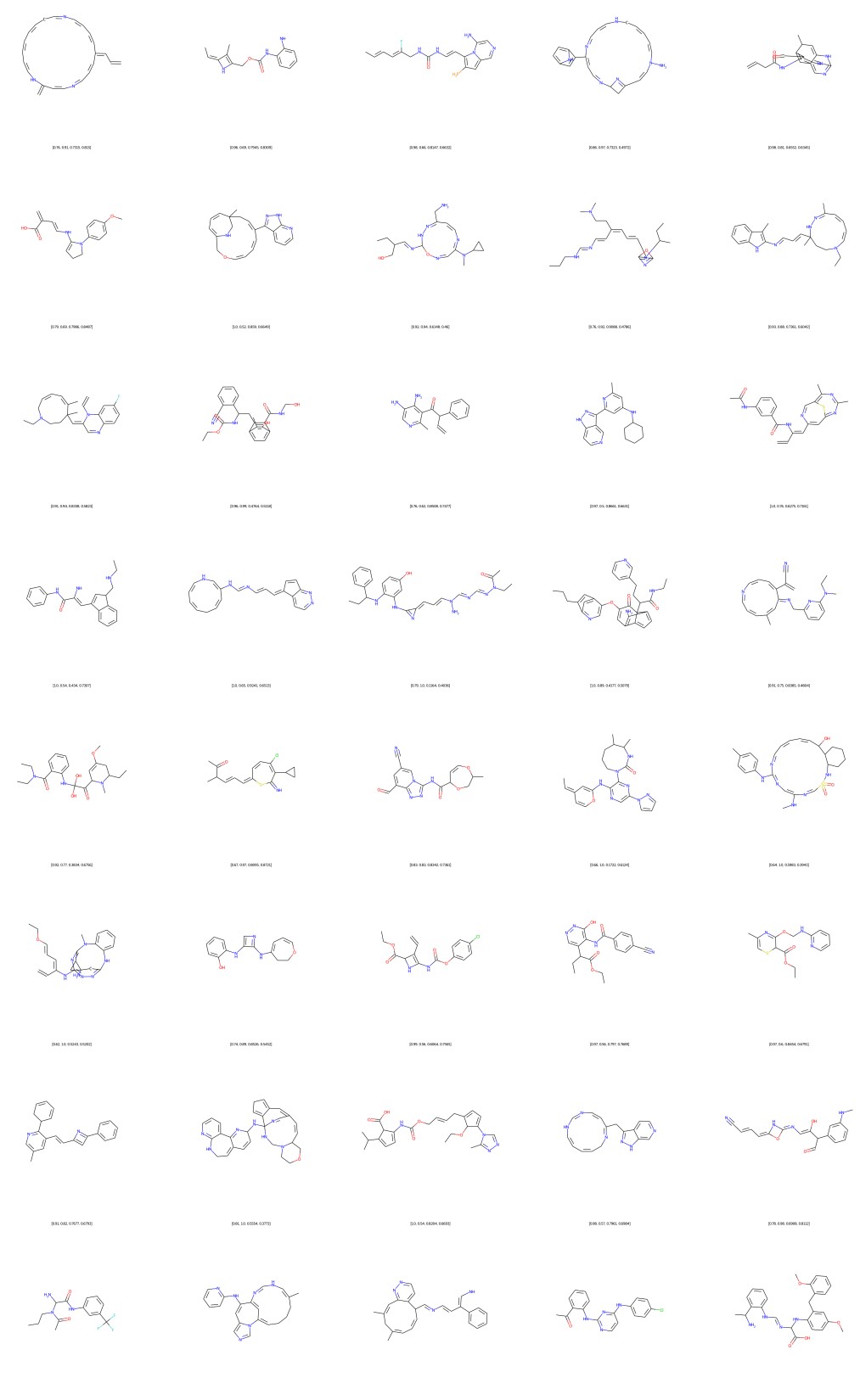

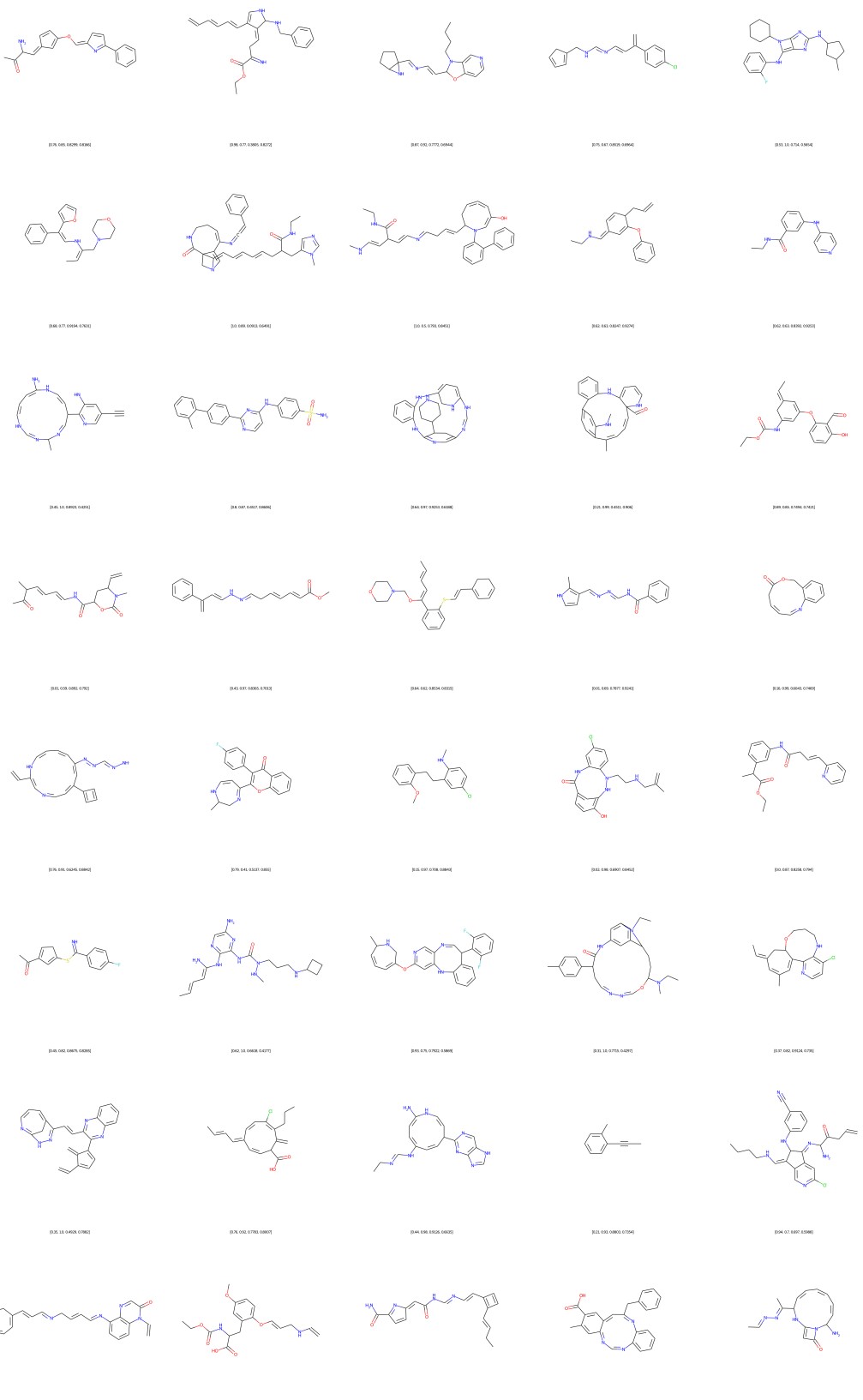

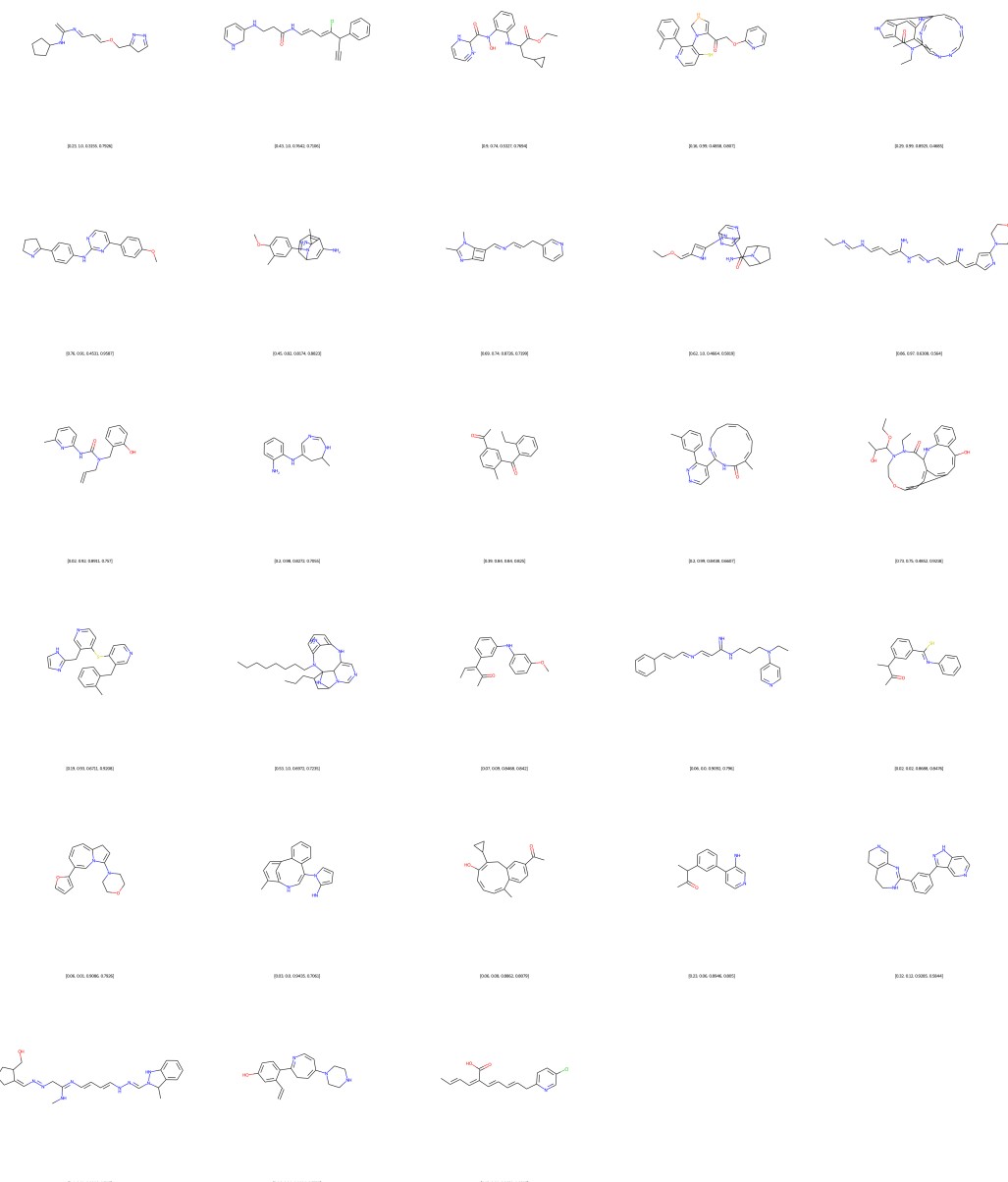

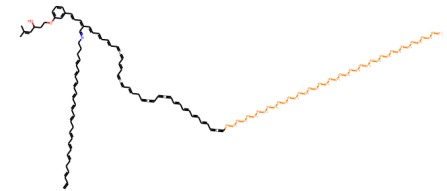

[0.89, 0.92, 0.7272, 0.726]

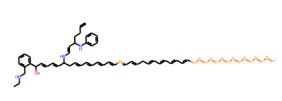

[0.18, 0.99, 0.8506, 0.6581]

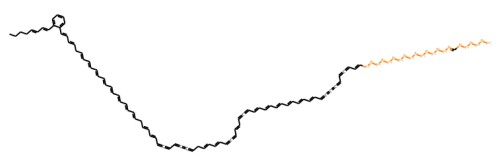

[0.79, 0.71, 0.779, 0.738]

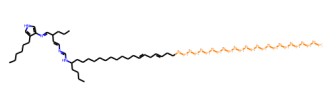

[0.77, 0.94, 0.8665, 0.4176]

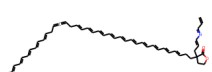

[0.1, 0.86, 0.8308, 0.773]

