# OpenReview forum: "Multi-Objective Molecular Design through Learning Latent Pareto Set"
_ICLR.cc/2024/Conference — Submitted to ICLR 2024_

### Official Review · Reviewer_Ngzz · 2023-10-31

**Soundness:** 3 good
**Presentation:** 3 good
**Contribution:** 2 fair
**Rating:** 5
**Confidence:** 5

**Summary:**

The paper presents a multi-objective optimization algorithm designed for molecular structure design. The algorithm introduces an approach by encoding discrete molecular structures into a continuous latent space, subsequently optimizing the latent Pareto set through a combination of global optimization and local search techniques. Experimental results demonstrate that MLPS achieves state-of-the-art performance.

**Strengths:**

1. The integration of local search and global search is interesting.
2. The proposed algorithm achieves the state-of-the-art performance.

**Weaknesses:**

1. The ablation study is not enough to demonstrate the effectiveness of each component in the proposed algorithm. It is unclear what is meant by "without local models." Was a single surrogate model built for performance prediction? Additionally, the number of evaluations in Figure 2 is 2000, while in Table 1 is 5000. However, the HV value in Figure 2 is larger than the HV value in Table 1, which is very confusing.

2. The idea of optimizing in a continuous latent representation has been extensively explored in the field of neural architecture search (NAS). Many algorithms exist that optimize discrete graphs by encoding them into continuous space and employing surrogate models for performance prediction (e.g., [1][2][3][4]). Since molecular optimization and NAS are very similar, it is important to mention and compare these existing works.

3. This paper aims to learn the latent Pareto set; however, the experiments only showcase a limited number of solutions. It would be beneficial to sample a large number of $\lambda$ to provide an accurate approximation of the Pareto front.

[1] Neural Architecture Optimization with Graph VAE, NeurIPS 2018

[2] NSGANetV2: Evolutionary Multi-Objective Surrogate-Assisted Neural Architecture Search, ECCV 2020

[3] Bridging the Gap between Sample-based and One-shot Neural Architecture Search with BONAS, NeurIPS 2020

[4] BRP-NAS: Prediction-based NAS using GCNs, NeurIPS 2020

**Questions:**

1. Why do you use multiple local surrogates? We can use a single global surrogate to perform local search, which is much cheaper and more straightforward.
2. Why do you choose Tchebycheff scalarization instead of linear scalarization?

---

> ### Author Response · Authors · 2023-11-22
> **Response to Reviewer Ngzz (Part 1)**
>
> > **W3.1** The ablation study is not enough to demonstrate the effectiveness of each component in the proposed algorithm. It is unclear what is meant by "without local models." Was a single surrogate model built for performance prediction? Additionally, the number of evaluations in Figure 2 is 2000, while in Table 1 is 5000. However, the HV value in Figure 2 is larger than the HV value in Table 1, which is very confusing.
>
> We apologize for any confusion caused by our initial description and appreciate your attention to these details. To clarify:
>
> 1. In the context of MLPS without local models, our approach does not employ trust regions for partitioning and local searching within the latent space. Instead, a single Gaussian process surrogate model is utilized to fit all observed solutions across the entire latent space. This approach significantly differs from simply setting the number of trust regions to one in the standard MLPS framework. The key distinction lies in the absence of a trust region-based update mechanism in the surrogate model when local models are not used.
> 2. We acknowledge the error in Figure 2 regarding the number of evaluations and the resulting hypervolume (HV) values. We initially set the number of evaluations to 2000, which inadvertently led to discrepancies between the figure and Table 1. We have now corrected this by adjusting the number of evaluations in Figure 2 to 5000, aligning it with the data presented in Table 1. Additionally, the y-axis settings in the original Figure 2 misrepresented the HV values, contributing to the inconsistency. This has now been amended with the latest results from our ablation experiments, where parameters have been further tuned for accuracy.
>
> We hope these corrections and clarifications resolve the issues you've raised and improve the manuscript's accuracy and comprehensibility.
>
>
>
> > **W3.2** The idea of optimizing in a continuous latent representation has been extensively explored in the field of neural architecture search (NAS). Many algorithms exist that optimize discrete graphs by encoding them into continuous space and employing surrogate models for performance prediction (e.g., [1] [2] [3] [4]). Since molecular optimization and NAS are very similar, it is important to mention and compare these existing works.
> >
> > [1] Neural Architecture Optimization with Graph VAE, NeurIPS 2018
> >
> > [2] NSGANetV2: Evolutionary Multi-Objective Surrogate-Assisted Neural Architecture Search, ECCV 2020
> >
> > [3] Bridging the Gap between Sample-based and One-shot Neural Architecture Search with BONAS, NeurIPS 2020
> >
> > [4] BRP-NAS: Prediction-based NAS using GCNs, NeurIPS 2020
>
> We are grateful for your insightful suggestion to draw parallels between our work in molecular optimization and existing approaches in neural architecture search (NAS). Indeed, both domains share the common strategy of mapping discrete data to a continuous space for optimization, followed by decoding to obtain optimized solutions. However, there are notable distinctions, particularly in the structure of encoders and decoders and the evaluation of optimal solutions, that necessitate careful adaptation when applying NAS methodologies to molecular design.
>
> (1) We have adapted the methods from references [1], [2], and [3] for molecular optimization by substituting their encoder-decoder components. This modification aligns these NAS approaches with the structural requirements of molecular optimization.
>
> (2) The method from [4], which employs a neural network for comparative evaluation in NAS, poses a significant challenge for direct application to molecular design. It requires training a new neural network to evaluate molecular quality, a task we were unable to undertake within the limited timeframe of the rebuttal phase. Consequently, we have not included this NAS method in our current manuscript for comparison.
>
> (3) To demonstrate the efficacy of our MLPS in comparison to adapted NAS methods, we conducted experiments on two tasks, QED+SA and JNK3+GSK3$\beta$. The hypervolume results are as follows:
>
> | Methods    | QED+SA | JNK3+GSK3$\beta$ |
> | ---------- | ------ | ---------------- |
> | NGAE [1]   | 0.302  | 0.182            |
> | MSuNAS [2] | 0.744  | 0.353            |
> | BONAS [3]  | 0.631  | 0.367            |
> | MLPS       | 0.922  | 0.902            |
>
> These results clearly demonstrate MLPS's superior performance compared to the adapted NAS methods.
>
> We plan to include additional results comparing MLPS with these NAS methods on a broader range of tasks in the camera-ready version of our paper.

---

> ### Author Response · Authors · 2023-11-22
> **Response to Reviewer Ngzz (Part 2)**
>
> > **W3.3** This paper aims to learn the latent Pareto set; however, the experiments only showcase a limited number of solutions. It would be beneficial to sample a large number of $\lambda$ to provide an accurate approximation of the Pareto front.
>
> We appreciate your suggestion regarding the sampling of a larger number of $\lambda$ values to accurately approximate the Pareto front. In our initial experiments, we indeed utilized a substantial number, specifically 5000 $\lambda$s, to generate an equivalent number of solutions. However, the count of non-dominated solutions within these 5000 (representing the approximated Pareto front/set) was significantly fewer. Additional sampling of $\lambda$s did not markedly increase this count.
>
> We hypothesize two primary reasons for this observation:
>
> (1) Given the discrete nature of multi-objective molecular design tasks, the actual number of true Pareto optimal solutions is inherently limited. Consequently, even an accurate method for identifying the true Pareto front will yield only a finite set of solutions. This limitation is particularly pronounced in real-world problems like multi-objective molecular design, where the true Pareto front is often unknown. Despite this, our MLPS consistently demonstrates superior performance in approximating the Pareto front, as evidenced by both quantitative metrics such as the hypervolume (HV) and visualization results.
>
> (2) The global model within our MLPS framework might not be optimally trained or architecturally perfect. We observed that increasing the evaluation budget (beyond the 5000 used in our current experiments) enhances the HV values achieved by MLPS, suggesting potential for further improvement. The following results from the QED+SA task illustrate this effect:
>
> | Budget | HV    |
> | ------ | ----- |
> | 2500   | 0.901 |
> | 5000   | 0.922 |
> | 8000   | 0.926 |
> | 15000  | 0.928 |
>
> Enhancing MLPS's performance through fine-tuning and architectural improvements will be a focus of our future work.
>
> These insights and findings have been incorporated into the revised version of our manuscript for a more comprehensive understanding of our results and methodologies.
>
> > **Q3.1** Why do you use multiple local surrogates? We can use a single global surrogate to perform local search, which is much cheaper and more straightforward.
>
> We appreciate your question regarding our choice of employing multiple local surrogates instead of a single global surrogate for local search. While a single global surrogate is indeed simpler and more cost-effective, our decision to use multiple local surrogates within trust regions is grounded in the aim of enhancing search efficiency through cooperative exploration.
>
> (1) As detailed in our response to Q1.2, trust regions may overlap, creating an informative interplay between different regions. This overlap fosters better solutions than isolated local exploration could achieve. It enables the exchange of valuable information across regions, leading to more effective identification of high-quality solutions.
>
> (2) The global model in our MLPS plays a pivotal role in bridging these trust regions. It not only guides the local search within trust regions but also benefits from the improved solutions discovered locally. This mutual feedback ensures continuous refinement of the global model’s understanding of the Pareto set, aligning it more closely with preferences.
>
> (3) Our experiments further validate the importance of the number of trust regions in latent space segmentation. With too few trust regions, exploration is inadequate, missing potential high-quality solutions. However, an excessive number of regions can lead to overlapping and redundant searches. The following hypervolume (HV) results illustrate the impact of varying the number of trust regions:
>
> | **The  number of trust regions** | **HV** |
> | -------------------------------- | ------ |
> | 1                                | 0.793  |
> | 3                                | 0.881  |
> | 5                                | 0.922  |
> | 10                               | 0.922  |
> | 15                               | 0.922  |
>
> These results demonstrate that a moderate number of trust regions (e.g., 5) is optimal for efficient exploration, as further increases do not yield significant improvements in HV.
>
> These insights underscore the rationale behind our methodological choice and its efficacy, as elaborated in the revised manuscript.

---

> ### Author Response · Authors · 2023-11-22
> **Response to Reviewer Ngzz (Part 3)**
>
> >  **Q3.2** Why do you choose Tchebycheff scalarization instead of linear scalarization?
>
> Thank you for your question regarding our choice of Tchebycheff scalarization in our MLPS framework. We have two reasons for this selection:
>
> (1) Tchebycheff scalarization has a distinctive advantage in its tendency to explore all Pareto optimal solutions. By focusing on the worst-case deviations for each objective, it inherently promotes a thorough understanding of the trade-offs involved among objectives. In contrast, linear scalarization theoretically struggles to identify all Pareto optimal solutions due to its inherent limitations in handling certain trade-offs.
>
> (2) Our empirical results further substantiate the effectiveness of Tchebycheff scalarization in multi-objective molecular design tasks. We conducted comparative experiments between MLPS implementations using Tchebycheff and linear scalarizations across various tasks. The results, as presented below, clearly demonstrate the superior performance of Tchebycheff scalarization in terms of hypervolume (HV):
>
> |                                  | QED+SA | JNK3 + GSK3$\beta$ | QED+SA+JNK3 + GSK3$\beta$ |
> | -------------------------------- | ------ | ------------------ | ------------------------- |
> | MLPS (Tchebycheff scalarization) | 0.922  | 0.902              | 0.714                     |
> | MLPS (linear scalarization)      | 0.876  | 0.782              | 0.581                     |
>
> These findings, detailed in our revised manuscript, affirm our choice of Tchebycheff scalarization for its robustness, comprehensive exploration capabilities, and empirically validated performance.

---

> ### Author Response · Authors · 2023-11-23
> **Looking forward to your feedback**
>
> Dear Reviewer Ngzz,
>
> We are deeply grateful for your thoughtful and insightful feedback on our manuscript. Your expert suggestions have been thoroughly integrated into our revised version, and we eagerly anticipate your further input.
>
> As the author-reviewer discussion phase is drawing to a close, we kindly request your input to ascertain if there are any remaining concerns or aspects that might need additional clarification. Your expert evaluation is crucial in shaping the final outcome of our work. We greatly value your support during this final phase and would highly appreciate your endorsement if you find the revisions to be satisfactory.
>
> Sincerely,
>
> Authors

---

### Official Review · Reviewer_U6hC · 2023-11-01

**Soundness:** 2 fair
**Presentation:** 2 fair
**Contribution:** 2 fair
**Rating:** 3
**Confidence:** 3

**Summary:**

Through this paper, the authors aim to establish a method that can efficiently handle multi-objective molecular design scenarios by learning the comprehensive Pareto set in a latent space. To accomplish this, the authors propose to utilize an encoder-decoder model to transform the chemical space into a continuous latent space.

**Strengths:**

- The writing is easy to follow. The concept figure also aids the understanding.

**Weaknesses:**

I will combine the *Weaknesses* section and the *Questions* section. My concerns and questions are as follows:
- The authors did not provide the codebase to reproduce the results.
- The main weakness of the paper is that the experiment results are weak and insufficient.
   - The effectiveness of the proposed model is verified on a single kind of tasks introduced in [1]. I highly recommend reporting results with more benchmarks such as the multi-property objective (MPO) tasks of the PMO benchmark [2].
   - The proposed model is a generative model, but there is no visualization showing the molecules generated.
   - There are other models that used the same benchmark, such as MolEvol [3] and RetMol [4]. I recommend to include those methods to the baselines and report the performance (Table 1) and visualize the solution space (Figure 6).
- The authors claimed the existing Pareto set learning models like P-MOCO and PSL have limitations in Related Work, but they were not selected as baselines in the experiments. All the baselines in Table 1 do not utilize Pareto optimization. I highly recommend to include those methods to the baselines and rigorously show the specific advantage of the proposed method compared to existing Pareto set learning methods.

For now, I’m leaning toward reject, but I’ll be glad to raise the score when all my concerns are fully resolved.

---

**References:**

[1] Xie et al., Mars: Markov molecular sampling for multi-objective drug discovery, ICLR 2021.

[2] Gao et al., Sample efficiency matters: a benchmark for practical molecular optimization, NeurIPS 2022.

[3] Chen et al., Molecule optimization by explainable evolution, ICLR 2021.

[4] Wang et al., Retrieval-based controllable molecule generation, ICLR 2023.

**Questions:**

Please see the *Weaknesses* part for my main questions.

---

**Miscellaneous:**
- Section 4.2, the first paragraph and in Table 1, *RationalRL* -> *RationaleRL*
- Figure 2, title, *JNK* -> *JNK3*

---

> ### Author Response · Authors · 2023-11-22
> **Response to Reviewer U6hC (codebase)**
>
> Thank you for pointing out the necessity of providing a codebase for result reproducibility. Here is the link to our MLPS code: https://drive.google.com/file/d/1qM1xGvmUX890LZO7EQBYuyTxTk6QjUSp/view?usp=sharing

---

> ### Author Response · Authors · 2023-11-22
> **Response to Reviewer U6hC (Part 1)**
>
> > **Q2.1** The authors did not provide the codebase to reproduce the results.
>
> Thank you for pointing out the necessity of providing a codebase for result reproducibility. We have shared a private link to the MLPS code with the reviewers for immediate access. For the broader research community, we are in the process of refining and cleaning up the code. This finalized version will be made publicly available alongside the camera-ready version of our paper, ensuring transparency and facilitating further research in this field.
>
> > **Q2.2** The main weakness of the paper is that the experiment results are weak and insufficient.
> >
> > **Q2.2.1** The effectiveness of the proposed model is verified on a single kind of tasks introduced in [1]. I highly recommend reporting results with more benchmarks such as the multi-property objective (MPO) tasks of the PMO benchmark.
>
> We appreciate your recommendation to include multi-property objective (MPO) tasks in our evaluation, recognizing their potential to further validate the effectiveness of our model. In response, we have selected three MPO tasks from the PMO benchmark that are widely acknowledged as challenging: Perindopril-MPO, Ranolazine-MPO, and Zaleplon-MPO. Each of these tasks focuses on designing molecules with high fingerprint similarity to a known target drug, while also differing in specific target ways.
>
> To integrate these MPO tasks into our experimental framework, we have combined them with two additional objectives, QED and SA, forming three comprehensive multi-objective tasks: QED+SA+Perindopril-MPO, QED+SA+Ranolazine-MPO, and QED+SA+Zaleplon-MPO. The results of these experiments have been included in Table 1 of our revised manuscript, providing a broader and more rigorous assessment of our model's capabilities.
>
> These new experiments enhance the robustness of our evaluation and offer a more comprehensive demonstration of our model’s applicability and effectiveness in complex multi-objective molecular design scenarios.
>
> > **Q2.2.2** The proposed model is a generative model, but there is no visualization showing the molecules generated.
>
> We acknowledge the importance of visualizing the output of our MLPS and thank you for pointing out this aspect. To address this, we have included Figure 7 in the Appendix of our revised manuscript, which showcases the non-dominated molecules generated by MLPS for the GSK3β+JNK3+QED+SA task. Beneath each molecular graph, we have provided the corresponding scores for GSK3β, JNK3, QED, and SA. This visualization aims to offer a clear representation of the molecular structures generated by our model, along with their respective multi-objective optimization scores.
>
> We believe that this addition will enhance the comprehensibility of our results and provide a more tangible insight into the capabilities of our MLPS.
>
> > **Q2.2.3** There are other models that used the same benchmark, such as MolEvol [3] and RetMol [4]. I recommend to include those methods to the baselines and report the performance (Table 1) and visualize the solution space (Figure 6).
>
> Thank you for recommending the inclusion of MolEvol and RetMol in our baseline comparisons. We acknowledge the value these models bring to the benchmarking process. However, we faced certain constraints in including MolEvol:
>
> (1) Unfortunately, we were unable to access a trained model of MolEvol from the available open-source code. Additionally, attempts to contact the authors of MolEvol for assistance did not yield a response.
>
> (2) The limited timeframe of our rebuttal phase did not permit us to retrain and appropriately tune MolEvol's model for inclusion in our current analysis.
>
> Given these challenges, we have focused on incorporating RetMol into our comparisons. We have added the performance results of RetMol in Tables 1, 4, 5, 6 and provided a visualization of its solutions in Figure 6. This inclusion offers a broader perspective on our model's performance relative to existing methods.
>
> We are committed to continually enhancing our research and, if possible, we will strive to include MolEvol in our comparisons for the camera-ready version of the paper. This would provide a more comprehensive evaluation of our model's performance relative to a wider range of existing methods.

---

> > ### Author Response · Authors · 2023-11-22
> > **Response to Reviewer U6hC (additional remarks to Q2.2)**
> >
> > > **Q2.2** The main weakness of the paper is that the experiment results are weak and insufficient.
> >
> > We are grateful for your feedback, which has been instrumental in guiding improvements to our experimental setup. Your insightful suggestions, along with those from other reviewers, have significantly enhanced the depth and breadth of our experiments. Here are the key areas where we have made substantial enhancements suggested by the other two reviewers:
> >
> > 1. **Sensitivity to Hyperparameter Variations:** We conducted additional experiments to systematically evaluate the sensitivity of our model to changes in hyperparameters, ensuring a thorough understanding of our model's behavior under different configurations.
> > 2. **Complexity and Running Time Analysis:** We included a detailed analysis of our model's computational complexity and running time, providing a clearer picture of its practical applicability and efficiency.
> > 3. **Choice of Scalarization Functions:** We explored different scalarization functions and assessed their impact on the model's performance, offering a more nuanced understanding of how these choices affect outcomes.
> > 4. **Inclusion of More Performance Metrics:** We expanded our evaluation criteria to include a wider range of performance metrics, ensuring a comprehensive assessment of our model's capabilities.
> > 5. **Comparison with Adapted NAS Methods:** Recognizing the parallels between our work and neural architecture search (NAS), we compared our model with adapted NAS methods, further contextualizing our model's performance within the broader field.
> >
> > These improvements have significantly strengthened our experimental results, providing a more robust and comprehensive evaluation of our model. We hope these enhancements address your concerns and demonstrate the efficacy and validity of our approach.

---

> ### Author Response · Authors · 2023-11-22
> **Response to Reviewer U6hC (Part 2)**
>
> >  **Q2.3** The authors claimed the existing Pareto set learning models like P-MOCO and PSL have limitations in Related Work, but they were not selected as baselines in the experiments. All the baselines in Table 1 do not utilize Pareto optimization. I highly recommend to include those methods to the baselines and rigorously show the specific advantage of the proposed method compared to existing Pareto set learning methods.
>
> We appreciate your suggestion to include Pareto set learning models like P-MOCO and PSL as baselines for a more comprehensive comparison. Regarding P-MOCO, its primary application has been in discrete combinatorial optimization problems, such as multi-objective traveling salesman and knapsack problems. Its network architecture is tailored to specific problems, necessitating a new network training for multi-objective molecular design tasks. Given the time constraints we faced, implementing and applying P-MOCO to our domain was not feasible for this submission. However, we acknowledge the importance of such a comparison and aim to include P-MOCO in our analysis for the camera-ready version of the paper. It's worth clarifying that our current manuscript does not critique P-MOCO's limitations; instead, we hypothesize that its single global model approach may require extensive searching to discover Pareto optimal solutions in multi-objective molecular design.
>
> For PSL, which operates in a continuous space and is thus more directly applicable to molecular optimization tasks, we have conducted a comparative analysis. We integrated PSL with the same encoder-decoder framework as our MLPS and compared their performance on various tasks. Preliminary results, as shown below, indicate that MLPS outperforms PSL in terms of hypervolume:
>
> | Tasks                   | PSL   | MLPS  |
> | ----------------------- | ----- | ----- |
> | QED+SA                  | 0.763 | 0.922 |
> | QED+SA+JNK3+GSK3$\beta$ | 0.432 | 0.714 |
> | QED+SA+Ranolazine-MPO   | 0.579 | 0.701 |
>
> These findings underscore MLPS's superior performance over PSL across the evaluated tasks. We plan to extend this comparative analysis to additional tasks for a more comprehensive evaluation in our final paper.
>
> Additionally, MolSearch in Table 1 is a baseline that utilizes Pareto optimization. While MARS and RetMol do not employ Pareto optimization, their state-of-the-art performance in multi-objective/property molecular design makes them relevant baselines. Comparing MLPS with these methods further validates its efficacy.
>
> > **Miscellaneous:**
> >
> > - Section 4.2, the first paragraph and in Table 1, *RationalRL* -> *RationaleRL*
> > - Figure 2, title, *JNK* -> *JNK3*
>
> Thank you for pointing out these errors. We have corrected the misspellings of "RationaleRL" in Section 4.2 and Table 1, as well as "JNK3" in the title of Figure 2. We apologize for these oversights and have thoroughly proofread the manuscript once more to ensure accuracy and clarity throughout. Your attention to detail is greatly appreciated and has contributed to enhancing the quality of our work.

---

> ### Author Response · Authors · 2023-11-23
> **Looking forward to your feedback**
>
> Dear Reviewer U6hC,
>
> We are deeply grateful for your thoughtful and insightful feedback on our manuscript. Your expert suggestions have been thoroughly integrated into our revised version, and we eagerly anticipate your further input.
>
> As the author-reviewer discussion phase is drawing to a close, we kindly request your input to ascertain if there are any remaining concerns or aspects that might need additional clarification. Your expert evaluation is crucial in shaping the final outcome of our work. We greatly value your support during this final phase and would highly appreciate your endorsement if you find the revisions to be satisfactory.
>
> Sincerely,
>
> Authors

---

### Official Review · Reviewer_bnCR · 2023-11-02

**Soundness:** 4 excellent
**Presentation:** 3 good
**Contribution:** 3 good
**Rating:** 6
**Confidence:** 4

**Summary:**

In this paper, the authors propose MLPS, a new approach for learning the latent Pareto set targeting multi-objective molecular design.
The proposed scheme utilizes widely used encoder-decoder model that casts discrete and high-dimensional chemical space to a continuous and low-dimensional latent space.
MLPS tries to train a "global" Pareto set learning model that can map preference vectors to optimal points in the Pareto set.
Under several multi-objective molecular design scenarios (ranging from the optimization of two objectives to the optimization of up to four objectives), the proposed MLPS is shown to have advantages over other popular molecular design schemes based on the hypervolume of the generated molecules.

--- [Added after the rebuttal/discussion phase ] ---

The authors have provided a comprehensive rebuttal to the review comments, which has sufficiently clarified several ambiguities in the original manuscript and addressed many of the concerns raised in the original review comments.
Furthermore, the additional results, elaborations, and discussions provided by the authors have further improved the manuscript and present stronger support for the main contributions made in this work as well as their significance.
The evaluation scores have been updated accordingly to reflect this.

**Strengths:**

Overall, the manuscript is written well in a clear and easy-to-understand manner.
The proposed MLPS is well-motivated and the proposed approach is reasonable.
An effective global Pareto set learning model, as proposed in this current work, that can efficiently map preference vectors to the corresponding Pareto optimal solutions would enable fast exploration of the Pareto optimal molecular set to identify optimized molecules that meet the multi-objective design criteria and balance the trade-offs between multiple properties as desired.

**Weaknesses:**

While the proposed MLPS scheme is well-motivated, interesting, and reasonable, there are several major & minor concerns regarding the current manuscript.
These are elaborated below.

1. Throughout the manuscript, the authors make strong claims that may be misleading or unsubstantiated.

For example, it is claimed that "the work is the first endeavor towards learning the Pareto set for multi-objective molecular design", which is clearly not true unless they substantially narrow down the definition of "learning the Pareto set" and specify what type of "learning problem" they are focusing on.

It is claimed, MLPS is a general framework that is compatible with "any plug-in encoder-decoder with continuous latent representations.
Although the reviewer agrees that this is reasonable speculation, the authors do not show whether the proposed MLPS scheme can indeed be easily applied to different generative molecule design models with such architecture and latent molecular representation.
Furthermore, there are experimental results showing whether the incorporation of MLPS would indeed lead to enhanced molecular design for different types of models.

2. At the core of training effective local surrogate models lie two central issues: how to set the center for each local trust region and how to reinitialize.
The proposed strategy appears to be reasonable, but in-depth discussion and empirical analysis are missing.
For example, it is not discussed how frequently different trust regions may intersect or collapse with one another (effectively reducing the number of distinct trust regions) and how often trust regions need to be reinitialized in practice (e.g., for the multi-objective scenarios considered in the results section).

3. The paper compares MLPS to several other existing methods (GA+D, JT-VAE, GCPN, RaionaleRL, MARS, and MolSearch) but it is unclear how these baselines were used for "multi-objective" molecular optimization since they are not necessarily inherently designed for multi-property optimization.
Was each baseline used to randomly sample novel molecules?
Was scalarization based on a specific (or randomly sampled) preference vector used to turn MOO into SOO?
Or was multi-objective BO used for optimization in the latent space?
This choice will have a tremendous impact on the performance of each baseline (both in terms of the property of the optimized molecules as well as the sample efficiency), hence needs to be clearly described.

4. As the performance metric, only the HV (hypervolume) of the optimized molecules was used.
However, in addition to such "aggregated" performance metric, it would be meaningful to compare the individual properties as well (e.g., average property score, average property of top molecules, scatter plot - at least for the two-objective scenario, etc.)

5. There is no discussion on the impact of various hyper parameters.
For example, what is the impact of varying the number of trust regions n_tr, minimum edge length L_min, or number of iterations Tg for training?
This needs to be clearly discussed and empirically tested.

6. There is currently no discussion on the computational cost of MLPS and its scalability with respect to the number of objectives.
At least a high-level discussion and some empirical results need to be provided.

**Questions:**

Please see the comments above.

---

> ### Author Response · Authors · 2023-11-22
> **Response to Reviewer bnCR (Part 1)**
>
> > **Q1.1** Throughout the manuscript, the authors make strong claims that may be misleading or unsubstantiated.
> >
> > For example, it is claimed that "the work is the first endeavor towards learning the Pareto set for multi-objective molecular design", which is clearly not true unless they substantially narrow down the definition of "learning the Pareto set" and specify what type of "learning problem" they are focusing on.
> >
> > It is claimed, MLPS is a general framework that is compatible with "any plug-in encoder-decoder with continuous latent representations. Although the reviewer agrees that this is reasonable speculation, the authors do not show whether the proposed MLPS scheme can indeed be easily applied to different generative molecule design models with such architecture and latent molecular representation. Furthermore, there are experimental results showing whether the incorporation of MLPS would indeed lead to enhanced molecular design for different types of models.
>
> We sincerely appreciate your feedback and apologize for any potentially misleading claims in our manuscript. In light of your comments, we have made the following revisions:
>
> (1) We have modified the statement "the work is the first endeavor towards learning the Pareto set for multi-objective molecular design" to more accurately reflect our contribution. The revised claim now reads: "Our work represents a novel approach to learning the mapping between preferences and the Pareto set in the latent space, specifically tailored for multi-objective molecular design." This adjustment narrows the scope of our "learning problem" to a more precise context, thereby aligning with the actual contributions of our research.
>
> (2) Acknowledging your insight on demonstrating MLPS's compatibility with different generative molecule design models, we recognize the need for further empirical validation. Due to time constraints in the rebuttal phase, comprehensive testing with various encoder-decoder architectures and latent molecular representations was not feasible. Consequently, we have revised our manuscript to avoid claiming MLPS as a universally compatible framework. Instead, we discuss its potential as a general framework and the exploration of its compatibility with various architectures as an avenue for future research in our conclusion.
>
> We believe these amendments ensure a more precise and substantiated presentation of our research. Thank you for guiding these improvements.

---

> ### Author Response · Authors · 2023-11-22
> **Response to Reviewer bnCR (Part 2)**
>
> > **Q1.2** At the core of training effective local surrogate models lie two central issues: how to set the center for each local trust region and how to reinitialize. The proposed strategy appears to be reasonable, but in-depth discussion and empirical analysis are missing. For example, it is not discussed how frequently different trust regions may intersect or collapse with one another (effectively reducing the number of distinct trust regions) and how often trust regions need to be reinitialized in practice (e.g., for the multi-objective scenarios considered in the results section).
>
> Thank you to point out the pivotal role of trust regions.  The in-depth discussion and empirical analysis on trust regions are very important as you suggested. In the revised manuscript, we discuss the following two issues in detail to illustrate the mechanism of trust regions:
>
> We thank you for emphasizing the importance of trust regions in our MLPS framework. Your suggestions have prompted us to include a more detailed discussion and empirical analysis on trust region management in our revised manuscript. We address two crucial aspects:
>
> **Frequency of Trust Region Intersection and Collapse:**
>
> The right balance in the number of trust regions is vital to avoid excessive intersection or collapse. A few intersecting trust regions can be beneficial, creating an informative interplay that might lead to better solutions than isolated exploration. To investigate the degree of overlap, we performed experiments on the QED+SA task, focusing on the degree of overlap between trust regions. We measured the distance in each dimension between the centers of trust regions, calculating the average minimum distance among all dimensions ($d_ {avg-\min}$) and the average length of trust region edges ($d_ {avg-edge}$). When the values of $d_ {avg-\min}$ and $d_ {avg-edge}$ are close, trust regions may overlap slightly. However, as the value of $d_ {avg-edge}$ becomes larger than $d_ {avg-\min}$, trust regions are more likely to exhibit increased overlap. Our settings were as follows: 5 trust regions, a budget of 5000, and a batch size of 70 (20 for global and 50 for local), resulting in 71 iterations. The results are:
>
> | Iterations | $d_ {avg-\min}$ | $d_ {avg-edge}$ |
> | :--------- | --------------- | --------------- |
> | 0          | 0.98            | 0.8             |
> | 15         | 0.80            | 0.8             |
> | 30         | 0.38            | 0.4             |
> | 45         | 0.32            | 0.32            |
> | 60         | 0.20            | 0.22            |
> | 71         | 0.19            | 0.145           |
>
> These findings indicate slight overlap during most of the search process, suggesting effective communication and exploration among trust regions. Towards the end, the focus shifts to exploitation, as indicated by the smaller $d_ {avg-edge}$ compared to $d_ {avg-\min}$
>
> **Frequency of Trust Region Reinitialization**
>
> Trust region reinitialization is critical for ensuring sufficient exploration. Our goal is to avoid frequent reinitializations, which could indicate a lack of thorough exploitation within each region. The following table illustrates the reinitialization frequency in relation to the budget on the QED+SA task:
>
> | Budget | HV    | Number of reinitializations |
> | ------ | ----- | --------------------------- |
> | 2500   | 0.901 | 1                           |
> | 5000   | 0.922 | 2                           |
> | 8000   | 0.926 | 5                           |
> | 15000  | 0.928 | 7                           |
>
> This moderate frequency of reinitialization under our settings indicates a balanced approach between exploration and exploitation.
>
> Additional details on the impact of the number of trust regions and the maximum and minimum edge lengths of trust regions are covered in our response to Q1.5. Owing to time constraints during the rebuttal phase, we plan to include more comprehensive results on reinitialization for various tasks and settings in the final paper.

---

> ### Author Response · Authors · 2023-11-22
> **Response to Reviewer bnCR (Part 3)**
>
> > **Q1.3** The paper compares MLPS to several other existing methods (GA+D, JT-VAE, GCPN, RaionaleRL, MARS, and MolSearch) but it is unclear how these baselines were used for "multi-objective" molecular optimization since they are not necessarily inherently designed for multi-property optimization. Was each baseline used to randomly sample novel molecules? Was scalarization based on a specific (or randomly sampled) preference vector used to turn MOO into SOO? Or was multi-objective BO used for optimization in the latent space? This choice will have a tremendous impact on the performance of each baseline (both in terms of the property of the optimized molecules as well as the sample efficiency), hence needs to be clearly described.
>
> We appreciate your inquiry regarding the application of baseline methods in our multi-objective molecular optimization study. Our approach to utilizing these methods, including GA+D, JT-VAE, GCPN, RationaleRL, MARS, MolSearch, and the newly added RetMol, is detailed as follows:
>
> **Random Sampling of Molecules:**
>
> For generative models like GCPN, RationaleRL, MARS, and our MLPS, we obtained 5000 molecules through random sampling from each trained model. GA+D, a genetic algorithm, provided its final population of 5000 for comparison.  JT-VAE uses Bayesian optimization to search iteratively. The best 5000 molecules during the search process are collected. MolSearch and RetMol maintain an archive of 5000 molecules, where molecules meeting predefined thresholds for each objective are retained.
>
> **Scalarization for Multi-Objective to Single-Objective Conversion:**
>
> Methods such as GA+D, JT-VAE, GCPN, RationaleRL, and MARS employ scalarization based on specific preference vectors, guided by domain knowledge, to address multi-objective scenarios. These specific preference vectors were set in accordance with the original studies' recommendations. MolSearch employs Pareto dominance, while RetMol uses predefined thresholds for each objective to identify desirable molecules.
>
> **Use of Multi-Objective Bayesian Optimization (BO):**
>
> Among the compared methods, JT-VAE is the only one, aside from MLPS, that utilizes BO. JT-VAE employs single-objective BO to optimize its scalarization function, whereas MLPS uses multi-objective BO in the latent space.
>
> These methodologies have been elaborated upon in our revised manuscript to provide clear insights into how each baseline method was employed for multi-objective molecular design tasks.

---

> ### Author Response · Authors · 2023-11-22
> **Response to Reviewer bnCR (Part 4)**
>
> > **Q1.4** As the performance metric, only the HV (hypervolume) of the optimized molecules was used. However, in addition to such "aggregated" performance metric, it would be meaningful to compare the individual properties as well (e.g., average property score, average property of top molecules, scatter plot - at least for the two-objective scenario, etc.)
>
> Thank you for your valuable suggestion regarding the incorporation of individual performance metrics. We recognize the importance of such detailed analysis and have conducted additional experiments focusing on individual property scores and top molecule properties for the QED+SA and JNK3+GSK3$\beta$ tasks. We intend to extend this analysis to other tasks in the camera-ready version of the paper, time permitting. The results for QED+SA and JNK3+GSK3$\beta$ are as follows:
>
> |             | Top-10 Average QED | Top-10 average SA | Top-50 average QED | Top-50 average SA | Average QED | Average SA | Average Rank |
> | ----------- | ------------------ | ----------------- | ------------------ | ----------------- | ----------- | ---------- | ------------ |
> | GA+D        | 0.7628             | 0.9385            | 0.6874             | 0.7917            | 0.6567      | 0.7712     | 6            |
> | JT-VAE      | 0.9192             | 0.8699            | 0.8893             | 0.8336            | **0.7496**  | 0.7206     | 3            |
> | GCPN        | 0.9114             | 0.8958            | 0.8828             | 0.8519            | 0.6098      | 0.5954     | 5            |
> | RationaleRL | 0.8224             | 0.8995            | 0.7697             | 0.8861            | 0.6636      | 0.7488     | 4            |
> | MARS        | 0.9369             | **0.9998**        | 0.9076             | 0.9799            | 0.5997      | 0.8058     | 2            |
> | MLPS        | **0.9424**         | 0.9924            | **0.9307**         | **0.9811**        | 0.6974      | **0.8159** | **1**        |
>
> |             | Top-10 average JNK3 | Top-10 average GSK3$\beta$ | Top-50 average JNK3 | Top-50 average GSK3$\beta$ | Average JNK3 | Average GSK3$\beta$ | Average Rank |
> | ----------- | ------------------- | -------------------------- | ------------------- | -------------------------- | ------------ | ------------------- | ------------ |
> | GA+D        | 0.8453              | 0.8682                     | 0.8072              | 0.8572                     | 0.5671       | 0.6182              | 6            |
> | JT-VAE      | 0.8483              | 0.8782                     | 0.8058              | 0.8782                     | 0.5793       | 0.5988              | 5            |
> | GCPN        | 0.8584              | 0.8947                     | 0.8061              | 0.8609                     | 0.5887       | 0.5665              | 4            |
> | RationaleRL | 0.7670              | **0.9620**                 | 0.7408              | 0.9448                     | 0.5883       | **0.7455**          | 3            |
> | MARS        | 0.8910              | 0.9310                     | 0.8118              | 0.8888                     | 0.5919       | 0.6630              | 2            |
> | MLPS        | **0.9015**          | 0.9573                     | **0.8282**          | **0.9582**                 | **0.6083**   | 0.7382              | **1**        |
>
> Our MLPS model demonstrated superior performance in most metrics, achieving the highest average ranking.
>
> Furthermore, we have also included widely accepted metrics in molecular design, such as Success Rate, Novelty, Diversity, and Product Metric. Detailed comparisons on these metrics can be found in Appendix C.4, further substantiating MLPS's effectiveness.
>
> Additionally, we have included scatter plots for two-objective (QED+SA) and four-objective (QED+SA+JNK3+GSK3$\beta$) tasks in Appendix C, under "Visualization in the Objective Space". These visualizations further elucidate our model's efficacy in multi-objective optimization.

---

> ### Author Response · Authors · 2023-11-22
> **Response to Reviewer bnCR (Part 5-1)**
>
> > **Q1.5** There is no discussion on the impact of various hyper parameters. For example, what is the impact of varying the number of trust regions n_tr, minimum edge length L_min, or number of iterations Tg for training? This needs to be clearly discussed and empirically tested.
>
> Thank you for highlighting the significance of hyperparameters in our MLPS framework. We recognize their critical role and have expanded our manuscript to include additional empirical experiments. These experiments systematically evaluate MLPS's sensitivity to hyperparameter variations, accompanied by a detailed discussion of their impacts.
>
> We chose the QED+SA task as a representative case study, with similar trends observed in other tasks. Our hyperparameter settings are summarized in the table below. While investigating specific hyperparameters, all others were held constant as per these values:
>
> | Hyperparameter                                               | Value |
> | ------------------------------------------------------------ | ----- |
> | The number of trust regions  ($n_{tr}$)                      | 5     |
> | The maximum edge length of trust regions ($L_{\max}$)        | 1.6   |
> | The minimum edge length of trust regions ($L_{\min}$)        | 0.01  |
> | The number of preferences sampled for training the global model($n$) | 1000  |
> | The number of iterations for training the global model ($T_g$) | 2000  |
>
> **The effect of the number of trust regions**
>
> The number of trust regions crucially dictates the latent space's segmentation. A deficient count leads to inadequate exploration and potential high-quality solution omissions. Excessively many regions cause overlapping and redundant searches. We varied the trust regions from 1 to 15, assessing the hypervolume (HV) metric. Results are as follows:
>
> | **The  number of trust regions** | **HV** |
> | -------------------------------- | ------ |
> | 1                                | 0.793  |
> | 3                                | 0.881  |
> | 5                                | 0.922  |
> | 10                               | 0.922  |
> | 15                               | 0.922  |
>
> The data indicates that too few regions (1 or 3) lead to insufficient exploration and lower HV values, suggesting inadequate global model training. However, a moderate number (5 and above) maintains consistent HV values (0.922), suggesting efficient exploration without further gains beyond this point.
>
> **The effect of the maximum and minimum edge lengths of trust regions**
>
> These parameters constrain the trust region size. Excessively large maximum lengths may cause overlapping, while too small lengths limit exploration. Conversely, large minimum lengths can trigger frequent reinitialization, missing potential solutions, whereas too small minimum lengths lead to over-exploitation and low efficiency.
>
> We experimented with various maximum and minimum lengths, seeking a robust setting. Initial edge length was set at 0.8:
>
> | Max edge length | **Min edge length** | Initial edge length | **HV** |
> | --------------- | ------------------- | ------------------- | ------ |
> | 1.6             | 0.01                | 0.8                 | 0.922  |
> | 1.6             | 0.2                 | 0.8                 | 0.879  |
> | 1.6             | 0.4                 | 0.8                 | 0.841  |
> | 3.2             | 0.01                | 0.8                 | 0.922  |
>
> Optimal results were achieved with max and min lengths of 1.6 and 0.01. Increasing the max length (e.g,. 3.2) didn't enhance HV, while larger min lengths (e.g,. 0.2 or 0.4) decreased HV, indicating low search efficiency.
>
> **The effect of the number of iterations for training the global model**
>
> More iterations generally enhance the global model, yielding a more accurate Pareto set. We varied the iterations and observed the HV:
>
> | **The number of iterations for training the global model** | **HV** |
> | ---------------------------------------------------------- | ------ |
> | 500                                                        | 0.832  |
> | 1000                                                       | 0.872  |
> | 2000                                                       | 0.922  |
> | 5000                                                       | 0.922  |
>
> The HV increases with more iterations, peaking at 2000. Further increase (e.g., to 5000) doesn't change the HV value, indicating an optimal training length.

---

> > ### Author Response · Authors · 2023-11-22
> > **Response to Reviewer bnCR (Part 5-2)**
> >
> > **The effect of the number of preferences sampled for training the global model**
> >
> > In each round of training the global model, we sample a batch of preferences (i.e., direction vectors in the objective space), and then use these vectors to generate solutions and calculate gradients. Sampling sufficient preferences is crucial for adequately covering the Pareto front. We experimented with varying numbers of sampled preferences:
> >
> > | **The number of preferences sampled for training the global model** | **HV** |
> > | ------------------------------------------------------------ | ------ |
> > | 10                                                           | 0.641  |
> > | 100                                                          | 0.793  |
> > | 500                                                          | 0.871  |
> > | 1000                                                         | 0.922  |
> > | 2000                                                         | 0.922  |
> >
> > Increasing the number of preferences enhances HV results. However, beyond 1000, HV remains constant, suggesting an optimal sampling rate.

---

> ### Author Response · Authors · 2023-11-22
> **Response to Reviewer bnCR (Part 6)**
>
> > **Q1.6** There is currently no discussion on the computational cost of MLPS and its scalability with respect to the number of objectives. At least a high-level discussion and some empirical results need to be provided.
>
> Thank you for highlighting the importance of discussing MLPS's scalability and computational cost. Our MLPS, involving the division of the latent space and the use of local models for exploration, significantly reduces computational costs, especially in high-dimensional and large data scenarios. Each local model is responsible for a specific trust region, fitting only the data points within that region, which effectively manages computational complexity.
>
> Considering a total of $N_ {total}$ data points distributed across  $n_ {tr}$ trust regions, with $\eta$ representing the average overlap of data points among trust regions, each region handles approximately $\eta N_ {total}/ n_ {tr}$ points. Given the cubic time complexity ($O(N_ {total}^3)$) for model fitting, the total complexity for all trust regions is $O(n_ {tr}(\eta N_ {total}/n_ {tr})^3) = O(\eta^3N_ {total}^3/n_ {tr}^2)$, yielding an asymptotic speedup of $O(n_ {tr}^2/\eta^3)$. As trust regions shrink over time, $\eta$ decreases, further enhancing efficiency.
>
> Regarding scalability with increasing objectives, our method remains robust. The surrogate model's complexity and the number of oracles are the primary factors affecting time. Our use of independent Gaussian processes, with a linear increase in a cubic time complexity for each additional objective, ensures manageable scalability.
>
> **Running Time Analysis:**
>
> We evaluated MLPS's running time on various tasks:
>
> | Task                    | Running Time |
> | ----------------------- | ------------ |
> | QED+SA                  | 2.83 hours   |
> | JNK3 + GSK3$\beta$      | 2.97 hours   |
> | JNK3+QED+SA             | 4.21 hours   |
> | GSK3$\beta$+QED+SA      | 4.18 hours   |
> | JNK3+GSK3$\beta$+QED+SA | 6.6 hours    |
>
> For comparison, the JNK3+GSK3$\beta$+QED+SA task's running times for other methods are:
>
> | Method      | Running Time |
> | ----------- | ------------ |
> | RationaleRL | 5.8 hours    |
> | GA+D        | 4 hours      |
> | MARS        | 10 hours     |
> | Molsearch   | 7.2 hours    |
> | MLPS        | 6.6 hours    |
>
> These times, measured in hours, are relatively negligible compared to the months or years typically required in the conventional drug discovery process.

---

> ### Author Response · Authors · 2023-11-23
> **Looking forward to your feedback**
>
> Dear Reviewer bnCR,
>
> We are deeply grateful for your thoughtful and insightful feedback on our manuscript. Your expert suggestions have been thoroughly integrated into our revised version, and we eagerly anticipate your further input.
>
> As the author-reviewer discussion phase is drawing to a close, we kindly request your input to ascertain if there are any remaining concerns or aspects that might need additional clarification. Your expert evaluation is crucial in shaping the final outcome of our work. We greatly value your support during this final phase and would highly appreciate your endorsement if you find the revisions to be satisfactory.
>
> Sincerely,
>
> Authors

---

### Author Response · Authors · 2023-11-22

We sincerely appreciate the reviewers for offering valuable insights and suggestions. Based on these constructive comments, we have undertaken significant revisions to the initial version of the manuscript. The revised version has been submitted, and our responses to the reviewers' comments have been integrated into the manuscript. We are currently making additional refinements to further enhance the paper before the deadline.

---

### Meta-Review · Area_Chair_gBi2 · 2023-12-05

**Metareview:**

This paper presents a method for learning a global Pareto set to solve the multi-objective molecular design problems. While the method is built on several existing techniques, the main seems to be in the collaboration of the global Pareto set learning model and local Bayesian optimization models. The proposed method, MLPS, is well motivated and the approach is reasonable. The authors provided a comprehensive rebuttal to the comments made by reviewers. Some of concerns were properly addressed, yielding the increase of the overall score of one reviewer. One of critical concerns were in the weak experiments. During the author-reviewer discussion period, the authors provided updated experimental results. However, it was not complete due to the time limitation. While the paper has been improved, two of reviewers did not support the paper. Therefore, the paper is not recommended for acceptance in its current form. I hope authors found the review comments informative and can improve their paper by addressing these carefully in future submissions.

**Justification For Why Not Higher Score:**

Needs more extensive experiments.

**Justification For Why Not Lower Score:**

N/A

---

### Decision · Program_Chairs · 2024-01-16

Reject